# Two-Way Is Better Than One: Bidirectional Alignment with Cycle Consistency for Exemplar-Free Class-Incremental Learning

**Hongye Xu**
Chester F. Carlson Center for Imaging Science
Rochester Institute of Technology
`hx5239@rit.edu`

**Bartosz Krawczyk**
Chester F. Carlson Center for Imaging Science
Rochester Institute of Technology
`bartosz.krawczyk@rit.edu`

## Abstract

Continual learning (CL) seeks models that acquire new skills without erasing prior knowledge. In exemplar-free class-incremental learning (EFCIL), this challenge is amplified because past data cannot be stored, making representation drift for old classes particularly harmful. Prototype-based EFCIL is attractive for its efficiency, yet prototypes drift as the embedding space evolves; thus, projection-based drift compensation has become a popular remedy. We show, however, that existing one-directional projections introduce systematic bias: they either retroactively distort the current feature geometry or align past classes only locally, leaving cycle inconsistencies that accumulate across tasks. We introduce **BiCyc**, a **bi**directional projector alignment approach with a **cyc**le-consistency objective: two maps, old→new and new→old, are optimized with stop-gradient gating so that transport and representation co-evolve. Analytically, we prove that the cycle loss contracts the singular spectrum toward unity in whitened space and that improved transport of class means/covariances yields smaller perturbations of classification log-odds, preserving old-class decisions and directly mitigating catastrophic forgetting. Empirically, across standard EFCIL benchmarks, our method substantially reduces forgetting and improves accuracy in from-scratch settings, while remaining competitive in the pretrained fine-grained regime. The code is available at `https://github.com/HXuSz11/BiCyc_ICLR2026`.

## 1 Introduction

Continual learning (CL) studies models that learn from a stream of tasks without retraining from scratch or erasing prior knowledge (Parisi et al., 2019; Lange et al., 2022; Zenke et al., 2017). A widely used protocol is *class-incremental learning* (CIL), where tasks introduce disjoint labels and the learner must recognize all seen classes at test time without task identifiers. While rehearsal with stored exemplars often curbs forgetting (Lopez-Paz & Ranzato, 2017; Riemer et al., 2018; Pham et al., 2021; Caccia et al., 2021; Wang et al., 2022b; Yang et al., 2023), privacy or memory constraints motivate the *exemplar-free* variant (EFCIL), which prohibits retaining raw inputs. Among the many directions to mitigate forgetting (Zenke et al., 2017; Lopez-Paz & Ranzato, 2017; Schwarz et al., 2018; Aljundi et al., 2018; Riemer et al., 2018; Serra et al., 2018; Saha et al., 2020; Pham et al., 2021; Caccia et al., 2021; Deng et al., 2021; Cha et al., 2021; Wang et al., 2022a;b; Slim et al., 2022; Wang et al., 2023; Yang et al., 2023; Shi & Wang, 2023; Wang et al., 2024), prototype-based EFCIL has emerged as a compelling compromise: the model caches compact class statistics (means/covariances), and inference proceeds via nearest-prototype or Bayes scores—achieving strict no-memory operation with modest compute.

The core difficulty in prototype-based EFCIL is representation drift: as the backbone adapts to new tasks, the embedding geometry shifts and previously cached statistics become stale, biasing predictions toward recent classes. Existing EFCIL solutions to drift largely follow two routes that differ in how they balance stability and plasticity.

**Covariance and geometry modeling.** This route improves robustness by shaping the feature geometry or the decision metric, commonly keeping the backbone partially/fixed to limit drift.

FeTrIL (Petit et al., 2023) freezes the backbone and translates features to synthesize pseudo-features for past classes, trading some plasticity for stability. FeCAM (Goswami et al., 2023) argues that Euclidean metrics are suboptimal under non-stationarity and adopts anisotropic (Mahalanobis) scoring with class-wise covariances, typically with a frozen extractor. PASS (Zhu et al., 2021) strengthens old-class representations via prototype augmentation and self-supervision without exemplars. These methods effectively mitigate forgetting by stabilizing or re-weighting the geometry, but they largely *avoid cross-space transport*; the price of stability is potentially limited adaptation to new tasks.

**Prototype drift compensation.** A second—and increasingly dominant—route retains backbone plasticity and explicitly *transports* outdated prototypes into the current space. SDC (Yu et al., 2020) projects new features toward the old space and updates old prototypes accordingly. ADC (Goswami et al., 2024) estimates drift adversarially by pushing new samples toward old prototypes, then "resurrects" past classes. LDC (Gomez-Villa et al., 2024) replaces hand-crafted updates with a learnable drift module that scales across regimes. EFC (Magistri et al., 2024) performs affinity-weighted, class-wise shifts that update prototypes in tandem with classifier training. AdaGauss (Rypeść et al., 2024) follows the learned-projector path but transports full Gaussian class statistics (means and covariances) into the new space for Bayesian inference rather than only moving class means. DPCR (He et al., 2025) mitigates representation drift by estimating

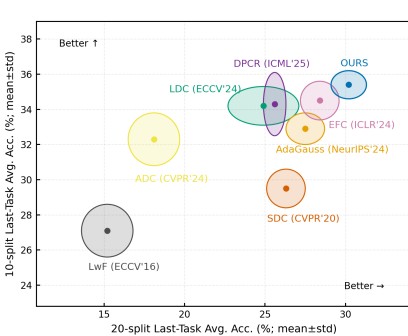

**Figure 1.** TinyImageNet ($T$=10): Our training algorithm yields solid performance gains over state-of-the-art EFCIL methods.

task-to-task semantic shift from stored class statistics and then analytically re-calibrating the classifier via ridge regression using only these cached moments, avoiding any exemplar replay. Despite strong performance, the prevailing paradigm here is *two-stage*: first train on the new task (often with regularization/distillation), then learn a post-hoc adapter (old→new). This paradigm leaves residual inconsistencies between spaces: transport is optimized only after the fact, and cycle errors accumulate over tasks.

**Our idea: from two-stage to *near* single-stage transport.** Motivated by the limitations of two-stage drift compensation, we propose bidirectional cycle consistency that evolves adapter training *into* the main task optimization so that transport and representation co-evolve. Concretely, during each new task we jointly learn two maps—$A : z_{\text{old}} \to z_{\text{new}}$ and $D : z_{\text{new}} \to z_{\text{old}}$—with *stop-gradient* targets to prevent retrograde updates on the evolving representation and a *cycle-consistency* objective that regularizes the pair toward a near-bijection on the data support. Analytically, we show that the cycle loss in whitened space equals $\|\tilde{A}\tilde{D} - I\|_F^2$ and contracts the singular spectrum of $\tilde{A}\tilde{D}$ toward one; and that smaller alignment/cycle errors yield tighter bounds on the perturbation of classification log-odds, preserving old-class decisions. After the main stage, a brief consolidation fine-tune is applied; inference uses a Gaussian Bayes classifier built from transported old-class statistics and freshly estimated current-task statistics.

**Contributions.**

- **Bidirectional cycle consistency (BiCyc) within training.** We formulate paired projections $A$ (old→new) and $D$ (new→old) learned *during* the task, with stop-gradient gating and a cycle loss that enforces near-inverse behavior on-support—addressing the asymmetry and post-hoc mismatch of prior two-stage, one-way pipelines.

- **Geometry-preserving transport for drift mitigation.** Our transport keeps old-class geometry stable as the backbone changes, yielding reduced recency bias and higher knowledge retention.

- **Theory-grounded alignment.** We prove that minimizing the cycle loss contracts the singular spectrum toward unity in whitened space and derive bounds linking mean/covariance transport errors to classification log-odds stability, explaining the observed reduction in forgetting.

- **Near single-stage pipeline with strong results.** By collapsing adapter learning into the main stage (with a short consolidation fine-tune), our method strikes an excellent balance

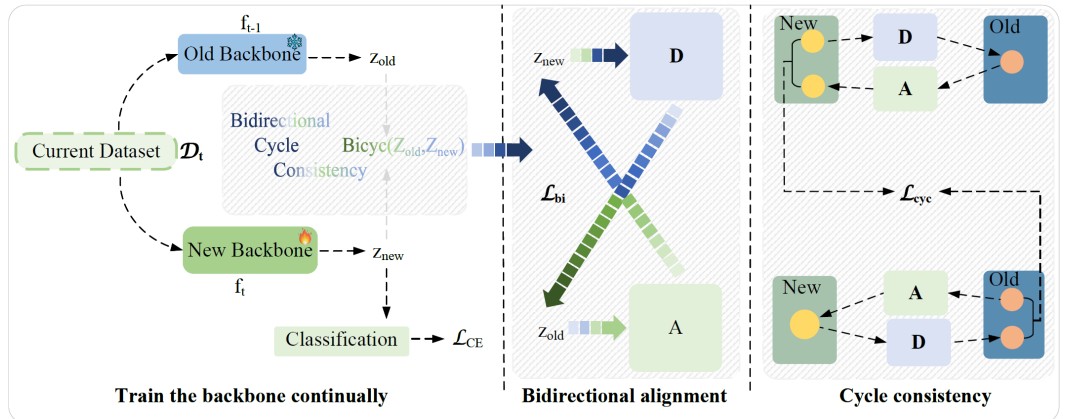

**Figure 2. Overview. (1) Train:** the current backbone $f_t$ learns on $\mathcal{D}_t$ (producing $z_{\text{new}}$, while frozen $f_{t-1}$ provides $z_{\text{old}}$) with task loss $\mathcal{L}_{\text{CE}}$. **(2) Bidirectional alignment:** jointly learn a distiller $D : z_{\text{new}} \to z_{\text{old}}$ and an adapter $A : z_{\text{old}} \to z_{\text{new}}$ using $\mathcal{L}_{\text{bi}}$. **(3) Cycle consistency:** enforce $A \circ D \approx I$ and $D \circ A \approx I$ with $\mathcal{L}_{\text{cyc}}$, yielding near-bijective, geometry-preserving transport. Old Gaussian prototypes are mapped forward by $A$, and all classes are evaluated in the *new* space.

between preserving stability (i.e., preventing drift) and maintaining plasticity, substantially reducing forgetting and maintaining or improving new-task accuracy across CIFAR-100, TinyImageNet, ImageNet-100, and CUB-200 under multiple splits. We discuss limitations in the experiments section.

## 2 PRELIMINARIES

### 2.1 PROBLEM DEFINITION

Continual learning (CL) aims to train a model on a stream of tasks while preserving previously acquired knowledge. In the **class-incremental** scenario considered here, each task $t \in \{1, \ldots, T\}$ introduces a disjoint label set $\mathcal{C}_t$ with $\mathcal{C}_i \cap \mathcal{C}_j = \varnothing$ for $i \neq j$. After learning task $t$, the model must recognize any class in $\mathcal{C}_{1:t} := \bigcup_{i=1}^{t} \mathcal{C}_i$ without a task identifier at test time.

Let $f_t : \mathcal{X} \to \mathbb{R}^d$ denote the feature extractor after completing the first $t$ tasks. During training on task $t$, the learner has access only to $\mathcal{D}_t = \{(x_i, y_i) \mid y_i \in \mathcal{C}_t\}$. The absence of any prior-task data defines the exemplar-free class-incremental setting.

### 2.2 PROTOTYPE-BASED EXEMPLAR-FREE CIL

In **exemplar-free class-incremental learning (EFCIL)**, the learner is prohibited from retaining raw samples from prior tasks. In the absence of replayed data, a common strategy is to summarize past knowledge with *prototypes*—one representative feature mean per seen class. Focusing on a single transition $t-1 \to t$, after completing task $t-1$ the learner stores for each class $c \in \mathcal{C}_{1:t-1}$ the prototype.

$$\boldsymbol{\mu}_c^{t-1} = \frac{1}{|\mathcal{D}_c|} \sum_{x \in \mathcal{D}_c} f_{t-1}(x), \tag{1}$$

where $\mathcal{D}_c$ collects all examples of class $c$ encountered up to step $t-1$. This summary is compact—its memory scales as $\mathcal{O}(|\mathcal{C}_{1:t-1}| \, d)$ for feature dimension $d$—and can be used at inference time either directly with a nearest-class-mean rule or to regularize subsequent training.

**Prototype drift.** When adapting the backbone from $f_{t-1}$ to $f_t$ on $\mathcal{D}_t$, the representation changes to fit the new classes and, as a side effect, the geometry of the feature space shifts. Hence, prototypes computed under $f_{t-1}$ become stale once $f_t$ is deployed. Denote the updated class mean, its vector displacement, and its norm by

$$\boldsymbol{\mu}_c^t = \frac{1}{|\mathcal{D}_c|} \sum_{x \in \mathcal{D}_c} f_t(x), \quad \boldsymbol{\Delta}_c^t = \boldsymbol{\mu}_c^t - \boldsymbol{\mu}_c^{t-1}, \quad \delta_c^t = \|\boldsymbol{\Delta}_c^t\|_2. \tag{2}$$

Larger $\delta_c^t$ indicates stronger prototype drift, which biases decisions toward recently learned classes.

Because no earlier samples are retained, $\boldsymbol{\mu}_c^t$ cannot be recomputed exactly; mitigating or compensating for this drift under the exemplar-free constraint motivates the two-stage strategy below.

## 2.3 Prior Drift Compensation Paradigm

A widely adopted recipe to handle prototype drift in EFCIL proceeds in two stages.

**Stage I: in-task regularization (backward alignment via $D$).** During task $t$, the old backbone $f_{t-1}$ is frozen and used as a teacher, while the current backbone $f_t$ is trained on the new data $\mathcal{D}_t$ as the student. Let $g$ denote the classifier head (shared or task-specific). For each $x \in \mathcal{D}_t$ we define $z_{\text{old}} = f_{t-1}(x)$, $z_{\text{new}} = f_t(x)$ and the corresponding logits $\ell_{\text{old}} = g(z_{\text{old}})$, $\ell_{\text{new}} = g(z_{\text{new}})$. The student is optimized with the usual cross-entropy on new labels and a distillation/regularization term that constrains either features or logits relative to the teacher:

$$\mathcal{L}_{\text{S1}} = \mathbb{E}_{(x,y) \in \mathcal{D}_t} \Big[ \text{CE}(\ell_{\text{new}}, y) \; + \; \lambda \, D(\phi_{\text{new}}(x), \, \phi_{\text{old}}(x)) \Big], \tag{3}$$

where $\phi$ is either $z$ (feature) or $\ell$ (logit), and $D$ stands for a distillation/regularization operator with $\lambda > 0$ balancing the terms. This stage constrains the update of $f_t$ using only $\mathcal{D}_t$, thereby limiting the growth of $\delta_c^t$ for past classes.

**Stage II: post-hoc prototype transport (adapter learning).** After training $f_t$, both $f_{t-1}$ and $f_t$ are frozen and an adapter $A$ is learned on $\mathcal{D}_t$ to map old features into the new space. Concretely, $A$ is fitted on paired features $(f_{t-1}(x), f_t(x))$ by minimizing

$$\min_A \; \mathbb{E}_{x \in \mathcal{D}_t} \| A\left(f_{t-1}(x)\right) - f_t(x) \|_2^2, \tag{4}$$

with $A$ instantiated as a global translation operator, a class-conditioned translation, or a learnable MLP/linear projector (details vary across works; see Appendix A.1.1). Once trained, $A$ is applied to the cached prototypes from prior steps to relocate them into the current feature space:

$$\tilde{\boldsymbol{\mu}}_c^t \; = \; A\left(\boldsymbol{\mu}_c^{t-1}\right), \qquad c \in \mathcal{C}_{1:t-1}. \tag{5}$$

These transported prototypes $\{\tilde{\boldsymbol{\mu}}_c^t\}$ are then used by the classifier at inference under $f_t$, effectively compensating for the shift induced by the update from $f_{t-1}$ to $f_t$.

**Our Research Objective.** In the two-stage paradigm, the regularization term in Stage I (often a distillation loss) pulls the new encoder $f_t$ toward the frozen teacher $f_{t-1}$, whereas the Stage II adapter transports old prototypes forward from the space of $f_{t-1}$ to that of $f_t$. Functionally, these two modules act in opposite directions; structurally, a prior work (Rypeść et al., 2024) instantiates the distiller with *the same architecture* as the adapter but applies it in the reverse direction ($z_{\text{new}} \rightarrow z_{\text{old}}$ vs. $z_{\text{old}} \rightarrow z_{\text{new}}$). Our objective is to make this duality explicit already in Stage I: we co-learn a forward adapter $A$ and a backward distiller $D_t$ during Stage I, enforcing *bidirectional alignment and cycle consistency* in both function (mutual inverses on features) and structure (mirrored/tied parameters), so that prototype transport becomes more accurate by design.

## 3 Methodology

### 3.1 Setup

Let $f_{t-1}$ be the frozen old backbone from task $t-1$ and $f_t$ the backbone being trained at task $t$. For an input $x$,

$$z_{\text{old}} = f_{t-1}(x) \in \mathbb{R}^d, \qquad z_{\text{new}} = f_t(x) \in \mathbb{R}^d.$$

Unless otherwise noted, evaluation is performed *in the new feature space* of $f_t$ using a Bayes classifier (see Appendix A.1.2), with class statistics estimated from $\mathcal{D}_t$ (new classes) or transported into the new space (old classes). We instantiate two lightweight maps: a **distiller** $D : \mathbb{R}^d \rightarrow \mathbb{R}^d$ (new→old) and an **adapter** $A : \mathbb{R}^d \rightarrow \mathbb{R}^d$ (old→new), implemented as linear layers or shallow MLPs. We use the stop-gradient operator $\text{stopgrad}(\cdot)$ throughout.

### 3.2 Joint Training with Bidirectional Cycle Consistency

We train $f_t$, $A$, and $D$ jointly on $\mathcal{D}_t$, combining standard classification with teacher–student regularization and our bidirectional/cycle consistency. Let $g$ be the task-specific classifier head with logits $\ell_{\text{new}} = g(z_{\text{new}})$. For brevity, we denote the bidirectional alignment + consistency cycle module as **Bicyc**$(z_{\text{old}}, z_{\text{new}})$ (see Fig. 2).

**Bidirectional alignment.** We seek (i) *backward compatibility* by making $z_{\text{new}}$ projectable to the old space via $D$, and (ii) a *forward* map $A$ that transports old prototypes into the current space used for evaluation—without dragging $f_t$ backward. Concretely,

$$\mathcal{L}_{\text{bi}} = \|D(z_{\text{new}}) - z_{\text{old}}\|_2^2 + \|A(z_{\text{old}}) - \text{stopgrad}(z_{\text{new}})\|_2^2. \tag{6}$$

The first term updates $f_t$ and $D$ (feature-level distillation, new→old). The second term updates $A$ only (detached target), so $A$ *chases* the evolving new space (old→new) without reducing the plasticity of $f_t$. In a linear–Gaussian view, minimizing equation 6 reduces transport errors $\varepsilon_{\text{old}\to\text{new}} = \mathbb{E}\|A(z_{\text{old}}) - z_{\text{new}}\|^2$ and $\varepsilon_{\text{new}\to\text{old}} = \mathbb{E}\|D(z_{\text{new}}) - z_{\text{old}}\|^2$, which bound prototype mean/covariance mismatch after transport and help control margin drift.

**Cycle consistency.** While $\mathcal{L}_{\text{bi}}$ aligns both directions, it does not by itself prevent degeneracies (e.g., rank loss in weakly correlated directions). We therefore add a cycle loss that nudges the compositions toward identity on the data support:

$$\mathcal{L}_{\text{cyc}} = \|A(D(z_{\text{new}})) - \text{stopgrad}(z_{\text{new}})\|_2^2 + \|D(A(z_{\text{old}})) - \text{stopgrad}(z_{\text{old}})\|_2^2. \tag{7}$$

Targets are detached, so $\mathcal{L}_{\text{cyc}}$ *stabilizes* $(A, D)$ without pulling $f_t$. Spectrally, enforcing $A \circ D \approx I$ and $D \circ A \approx I$ contracts the singular values of the composed transports toward 1, curbing rank/energy loss and promoting near-isometric geometry preservation. Thus $\mathcal{L}_{\text{bi}}$ lowers transport error (alignment) while $\mathcal{L}_{\text{cyc}}$ regularizes the transport *operators* (near-bijection); together they yield faithful prototype transport and empirically reduce forgetting without sacrificing plasticity. We denote the weighted sum of the bidirectional alignment and cycle-consistency losses by:

$$\textbf{Bicyc}(z_{\text{old}}, z_{\text{new}}) := \lambda_{\text{bi}} \mathcal{L}_{\text{bi}} + \lambda_{\text{cyc}} \mathcal{L}_{\text{cyc}} \tag{8}$$

We analyze the cycle objective under centered features and full-rank covariances on the data support, passing to whitened variables $\tilde{z}_{\text{old}} = \Sigma_{\text{old}}^{-1/2} z_{\text{old}}$ and $\tilde{z}_{\text{new}} = \Sigma_{\text{new}}^{-1/2} z_{\text{new}}$. In this space $\mathbb{E}[\tilde{z}_{\text{new}}\tilde{z}_{\text{new}}^\top] = I$, and the expected cycle error equals the squared Frobenius distance of $\tilde{A}\tilde{D}$ to $I$. We now state the resulting contraction property.

**Theorem 1 (Cycle contraction).** Let $\Sigma_{\text{old}} = \mathbb{E}[z_{\text{old}} z_{\text{old}}^\top]$ and $\Sigma_{\text{new}} = \mathbb{E}[z_{\text{new}} z_{\text{new}}^\top]$ be full-rank on the data support and define whitened variables $\tilde{z}_{\text{old}} = \Sigma_{\text{old}}^{-1/2} z_{\text{old}}$, $\tilde{z}_{\text{new}} = \Sigma_{\text{new}}^{-1/2} z_{\text{new}}$ with induced maps $\tilde{A} = \Sigma_{\text{new}}^{-1/2} A \Sigma_{\text{old}}^{1/2}$ and $\tilde{D} = \Sigma_{\text{old}}^{-1/2} D \Sigma_{\text{new}}^{1/2}$. Let $M := \tilde{A}\tilde{D} - I$. If the features are centered, then:

$$\mathbb{E} \|M\tilde{z}_{\text{new}}\|_2^2 = \|M\|_F^2. \tag{9}$$

By Mirsky/Hoffman–Wielandt (Horn & Johnson, 2013) $\sum_{k=1}^{d}(\sigma_k(\tilde{A}\tilde{D}) - 1)^2 \leq \|M\|_F^2$ and hence $\max_k |\sigma_k(\tilde{A}\tilde{D}) - 1| \leq \|M\|_F$. In particular, if $\|M\|_2 < 1$ then $1 - \|M\|_2 \leq \sigma_k(\tilde{A}\tilde{D}) \leq 1 + \|M\|_2$ and $\kappa(\tilde{A}\tilde{D}) \leq \frac{1+\|M\|_2}{1-\|M\|_2}$. Consequently, minimizing $\mathcal{L}_{\text{cyc}}$ drives the singular values of $\tilde{A}\tilde{D}$ toward 1 on the data support, preventing rank loss and preserving geometry. Proof in Appendix A.2.1.

**Corollary 2 (Decision stability for classification).** Let old-class statistics be transported as $\hat{\mu}_c^t = A\mu_c^{t-1}$ and (for linear $A$) $\hat{\Sigma}_c^t = A\Sigma_c^{t-1}A^\top$. Assume evaluation uses the Bayes rule with Gaussian class-conditionals $(\mu_c^t, \Sigma_c^t)$ and priors $\pi_c$, with log-scores $\ell_c(x)$ as in Appendix A.1.2. Define mean transport errors $\delta_c := \|\hat{\mu}_c^t - \mu_c^t\|_{(\Sigma_c^t)^{-1}}$. If the alignment error $\varepsilon_{\text{old}\to\text{new}}^2 = \mathbb{E}\|Az_{\text{old}} - z_{\text{new}}\|_2^2$ and the cycle error $\varepsilon_{\text{cyc,new}}^2 = \mathbb{E}\|ADz_{\text{new}} - z_{\text{new}}\|_2^2$ are small, then:

$$\delta_c \lesssim \sqrt{\varepsilon_{\text{old}\to\text{new}}^2}, \qquad \|\tilde{\Sigma}^t - \Sigma^t\|_2 \lesssim C_1 \sqrt{\varepsilon_{\text{old}\to\text{new}}^2} + C_2 \varepsilon_{\text{cyc,new}}. \tag{10}$$

For any class pair $(i, j)$ and any $x$, let $m_{ij}(x) := |\ell_i(x) - \ell_j(x)|$ be the Bayes margin. Then the induced change in log-odds satisfies $|(\hat{\ell}_i - \hat{\ell}_j) - (\ell_i - \ell_j)| \lesssim C_\mu(\delta_i + \delta_j) + C_\Sigma\|\hat{\Sigma}^t - \Sigma^t\|_2$. Consequently, if $C_\mu(\delta_i + \delta_j) + C_\Sigma\|\hat{\Sigma}^t - \Sigma^t\|_2 < m_{ij}(x)$, the Bayes decision between $i$ and $j$ at $x$ remains unchanged after transport. Proof in Appendix A.2.2.

**Pitfall of anti-collapse loss.** For features $z \in \mathbb{R}^{B \times S}$, let $\Sigma = \frac{1}{B-1}(z - \bar{z})^\top(z - \bar{z})$. The AdaGauss anti-collapse loss (Rypeść et al., 2024) is

$$\mathcal{L}_{\text{ac}} = -\frac{1}{S}\sum_{i=1}^{S}\min(\text{chol}(\Sigma)_{ii}, \beta). \tag{11}$$

Table 1: Average incremental ($A_{\text{inc}}$) and last-task average ($A_{\text{last}}$) accuracy (%, mean $\pm$ std. over five runs) on CIFAR-100 and TinyImageNet when training the feature extractor from scratch. Best results are **bold**.

| Method | CIFAR-100 | | | | TinyImageNet | | | |
| | $T$=10 | | $T$=20 | | $T$=10 | | $T$=20 | |
| | $A_{\text{last}}$ | $A_{\text{inc}}$ | $A_{\text{last}}$ | $A_{\text{inc}}$ | $A_{\text{last}}$ | $A_{\text{inc}}$ | $A_{\text{last}}$ | $A_{\text{inc}}$ |
|---|---|---|---|---|---|---|---|---|
| EWC | 30.9±1.9 | 50.4±1.7 | 17.0±1.6 | 34.2±2.1 | 18.5±1.8 | 34.3±2.3 | 11.3±1.9 | 26.8±2.5 |
| LwF$_{\text{ECCV16}}$ | 31.9±1.1 | 51.8±1.5 | 17.6±1.2 | 39.2±1.7 | 27.1±1.5 | 39.6±2.0 | 15.2±1.6 | 31.5±2.1 |
| SDC$_{\text{CVPR20}}$ | 40.6±0.9 | 56.2±1.3 | 32.3±1.0 | 46.6±1.4 | 29.5±1.1 | 43.8±1.5 | 26.3±1.2 | 40.6±1.7 |
| PASS$_{\text{CVPR21}}$ | 30.8±1.2 | 48.3±1.1 | 17.6±0.8 | 31.1±1.3 | 24.5±0.6 | 39.5±1.0 | 18.5±1.4 | 30.4±1.9 |
| FeTrIL$_{\text{WACV23}}$ | 34.9±0.5 | 51.2±1.1 | 23.3±1.4 | 37.9±1.2 | 31.0±0.9 | 45.3±1.8 | 25.9±1.2 | 39.9±1.2 |
| FeCAM$_{\text{NeurIPS23}}$ | 32.4±0.5 | 48.7±0.9 | 21.1±1.0 | 34.5±1.3 | 30.9±0.9 | 44.9±1.4 | 24.9±0.8 | 37.9±1.4 |
| EFC$_{\text{ICLR24}}$ | 43.5±0.8 | 58.1±1.2 | 32.4±0.9 | 47.0±1.3 | 34.5±1.1 | 47.9±1.5 | 28.4±1.2 | 42.1±1.6 |
| ADC$_{\text{CVPR24}}$ | 46.5±1.2 | 61.4±1.6 | 35.1±1.4 | 51.7±1.8 | 32.3±1.5 | 43.0±1.9 | 18.1±1.6 | 36.0±2.1 |
| LDC$_{\text{ECCV24}}$ | 45.4±1.6 | 59.5±1.9 | 35.5±1.9 | 51.9±2.3 | 34.2±1.1 | 46.8±1.6 | 24.9±2.2 | 38.2±2.7 |
| AdaGauss$_{\text{NeurIPS24}}$ | 46.8±1.2 | 60.9±1.0 | 37.9±1.0 | 54.4±0.8 | 32.9±0.9 | 45.8±1.3 | 27.5±1.2 | 39.5±1.1 |
| DPCR$_{\text{ICML2025}}$ | 50.2±0.7 | 62.8±1.1 | 39.8±1.2 | 54.8±0.9 | 34.3±1.8 | 46.9±0.9 | 25.6±0.7 | 39.3±0.6 |
| **BiCyc (Ours)** | **50.6±0.9** | **63.2±1.3** | **41.5±1.1** | **56.5±1.3** | **35.4±0.8** | **49.1±1.4** | **30.2±1.1** | **44.2±1.3** |

In practice, mini-batch $\Sigma$ can be non-SPD or rank-deficient, causing Cholesky failures and potentially inflating scale near ill-conditioning. We enforce SPD via symmetrization and shrinkage, with a jittered Cholesky and eigenvalue flooring as fallback:

$$\tilde{\Sigma} = \frac{1}{2}(\Sigma + \Sigma^{\top}), \qquad \hat{\Sigma} = \tilde{\Sigma} + \lambda \frac{\text{tr}(\tilde{\Sigma})}{S} I + \varepsilon I, \tag{12}$$

and, for very small batches, we optionally use a diagonal approximation $\hat{\Sigma}_{\text{diag}} = \text{diag}(\text{diag}(\hat{\Sigma}))$. The robust objective is

$$\mathcal{L}_{\text{ac}}^{\text{rob}} = -\frac{1}{S} \sum_{i=1}^{S} \min(\text{chol}(\hat{\Sigma})_{ii}, \beta). \tag{13}$$

**Total Stage-I loss and gradient routing.** Combining the classification, cycle, and anti-collapse terms yields:

$$\mathcal{L}_{\text{total}} = \underbrace{\mathcal{L}_{\text{CE}}(\ell_{\text{new}}, y)}_{\text{learn new classes}} + \textbf{Bicyc}(z_{\text{old}}, z_{\text{new}}) + \alpha \, \mathcal{L}_{\text{ac}}^{\text{rob}}. \tag{14}$$

Here, $\mathcal{L}_{\text{CE}}$ and the first term of equation 6 update $f_t$ (and $D$); the second term of equation 6 updates $A$ only (detached target); and equation 7 stabilizes $(A, D)$ without reducing the plasticity of $f_t$. Importantly, if gradients from the adapter are allowed to flow into $f_t$, $A$ and $D$ become adversarial, severely weakening $D$'s regularization and causing sharp performance drops. After Stage I, we freeze $f_{t-1}$, $f_t$, and $D$, and perform a low-learning-rate fine-tuning of $A$ on $\mathcal{D}_t$ to sharpen transport without re-optimizing from scratch.

## 4 EXPERIMENTS

**Baselines.** We benchmark our approach against a broad set of exemplar-free class-incremental learning (EFCIL) methods. Classic regularization baselines—EWC (Kirkpatrick et al., 2017) and LwF (Li & Hoiem, 2016)—are executed using the reference OCL implementation (Mai et al., 2022). Contemporary state-of-the-art approaches—SDC (Yu et al., 2020), PASS (Zhu et al., 2021), FeTrIL (Petit et al., 2023), FeCAM (Goswami et al., 2023), EFC (Magistri et al., 2024), ADC (Goswami et al., 2024), LDC (Gomez-Villa et al., 2024), and AdaGauss (Rypeść et al., 2024)—are run with the authors' public codebases as distributed via FACIL (Masana et al., 2023), PyCIL (Zhou et al., 2023), or the official repositories. Unless otherwise noted, we preserve the original data augmentations and default hyper-parameters reported by each paper.

**Implementation details and reproducibility.** We build on the public AdaGauss codebase and add the components introduced in this work. Unless stated otherwise, all experiments use a

Table 2: Average incremental ($A_{\text{inc}}$) and last-task average ($A_{\text{last}}$) accuracy (%, mean $\pm$ std. over five runs) on ImageNet-100 and CUB-200. Best results are **bold**. [†]: results excerpted from (Gomez-Villa et al., 2024). [‡]: results excerpted from (He et al., 2025).

| Method | ImageNet-100 | | | | CUB-200 | | | |
|---|---|---|---|---|---|---|---|---|
| | $T$=10 | | $T$=20 | | $T$=10 | | $T$=20 | |
| | $A_{\text{last}}$ | $A_{\text{inc}}$ | $A_{\text{last}}$ | $A_{\text{inc}}$ | $A_{\text{last}}$ | $A_{\text{inc}}$ | $A_{\text{last}}$ | $A_{\text{inc}}$ |
| EWC | 25.1±2.8 | 40.6±3.3 | 13.7±2.1 | 29.2±2.5 | 15.8±0.7 | 32.6±0.5 | 12.3±0.8 | 27.2±0.6 |
| LwF$_{\text{ECCV16}}$ | 33.4±2.2 | 51.5±1.6 | 18.6±1.6 | 41.3±1.9 | 30.4±1.1 | 46.1±1.0 | 19.4±1.6 | 34.7±1.8 |
| SDC$_{\text{CVPR20}}$ | 35.4±1.9 | 50.1±1.6 | 19.4±1.0 | 36.5±1.4 | 50.3±1.3 | 60.5±1.2 | 27.9±1.4 | 40.1±1.6 |
| PASS$_{\text{CVPR21}}$ | 26.4±1.3 | 45.7±0.2 | 14.4±1.2 | 31.7±0.4 | 27.0±0.9 | 42.3±0.9 | 18.1±1.2 | 36.9±1.1 |
| FeTrIL$_{\text{WACV23}}$ | 36.2±1.2 | 52.6±0.6 | 26.6±1.5 | 42.4±2.1 | 36.9±0.7 | 48.2±0.6 | 34.6±1.0 | 45.3±0.9 |
| FeCAM$_{\text{NeurIPS23}}$ | 38.7±1.0 | 54.8±0.5 | 29.0±1.3 | 44.6±2.0 | 40.2±0.8 | 54.9±1.0 | 36.2±1.1 | 48.9±1.3 |
| EFC$_{\text{ICLR24}}$ | 50.9±1.1 | 61.3±1.2 | 38.6±1.2 | 50.5±1.5 | 51.0±0.6 | 63.3±0.7 | **46.1±1.0** | **59.3±1.3** |
| ADC$_{\text{CVPR24}}$ | 38.3±1.2 | 55.5±1.5 | 25.1±1.3 | 43.4±1.7 | 49.5±0.9 | 58.8±1.1 | 35.4±1.4 | 48.3±1.4 |
| LDC$_{\text{ECCV24}}$ | 51.4[†]±1.2[†] | **69.4[†]±0.6[†]** | 28.5±1.7 | 46.5±2.7 | 47.5±0.7 | 55.7±1.3 | 27.2±1.1 | 39.8±2.1 |
| AdaGauss$_{\text{NeurIPS24}}$ | 51.1±1.2 | 65.0±1.4 | 42.6±1.6 | 57.4±1.9 | 52.9±0.8 | 63.4±1.3 | 45.0±1.3 | 57.0±1.0 |
| DPCR$_{\text{ICML2025}}$ | 49.9±0.8 | 64.8±1.1 | 37.3±1.6 | 54.7±0.7 | – | – | – | – |
| **BiCyc (Ours)** | **52.7±0.9** | 66.8±1.4 | **43.8±1.4** | **58.2±1.8** | **53.7±0.7** | **64.0±0.8** | 43.7±1.4 | 55.9±1.2 |

Table 3: Last-task average forgetting ($F_{\text{last}}$) (%, mean $\pm$ std. over five runs). CIFAR-100, TinyImageNet, and ImageNet-100 use from-scratch training; CUB-200 uses ImageNet-pretrained initialization. Best results are **bold**.

| Method | CIFAR-100 | | TinyImageNet | | ImageNet-100 | | CUB-200 | |
|---|---|---|---|---|---|---|---|---|
| | $T$=10 | $T$=20 | $T$=10 | $T$=20 | $T$=10 | $T$=20 | $T$=10 | $T$=20 |
| | $F_{\text{last}}$ | $F_{\text{last}}$ | $F_{\text{last}}$ | $F_{\text{last}}$ | $F_{\text{last}}$ | $F_{\text{last}}$ | $F_{\text{last}}$ | $F_{\text{last}}$ |
| LwF$_{\text{ECCV16}}$ | 23.2±1.7 | 31.2±1.8 | 21.9±1.9 | 33.5±2.4 | 42.1±2.3 | 48.1±2.2 | 16.5±1.1 | 21.7±1.4 |
| SDC$_{\text{CVPR20}}$ | 34.8±1.7 | 35.9±1.9 | 25.1±1.4 | 29.4±2.1 | 44.6±2.0 | 54.4±2.3 | 10.9±1.3 | 17.3±1.1 |
| EFC$_{\text{ICLR24}}$ | 23.1±1.1 | 24.7±1.8 | 23.5±2.4 | 30.1±3.0 | 21.5±1.9 | 23.8±2.5 | **10.7±0.7** | **14.8±1.7** |
| ADC$_{\text{CVPR24}}$ | 21.9±1.1 | 31.0±1.6 | 30.2±2.0 | 36.8±1.9 | 32.4±1.6 | 33.4±1.8 | 12.8±1.1 | 21.3±1.5 |
| LDC$_{\text{ECCV24}}$ | 21.7±1.9 | 25.6±2.3 | 24.7±2.5 | 30.7±2.1 | 25.7±1.7 | 32.9±2.3 | 13.6±1.2 | 23.9±1.8 |
| AdaGauss$_{\text{NeurIPS24}}$ | 16.7±1.4 | 21.0±1.5 | 18.7±1.2 | 23.1±1.0 | 20.6±0.9 | 22.9±1.1 | 11.6±0.7 | 16.9±1.3 |
| **BiCyc (Ours)** | **13.5±1.3** | **16.6±0.9** | **12.0±0.9** | **18.9±1.1** | **18.2±1.6** | **20.8±1.4** | 11.3±0.9 | 17.5±1.3 |

ResNet-18 backbone with a batch size of 256 images per iteration, following AdaGauss. For CIFAR-100 (Krizhevsky, 2009), TinyImageNet (Le & Yang, 2015), and ImageNet-100 (Deng et al., 2009), the backbone is trained from scratch for 200 epochs using SGD with a fixed initial learning rate of $1 \times 10^{-1}$ and weight decay $5 \times 10^{-4}$; the learning rate is decayed by a factor of 10 at epochs $\{60, 120, 180\}$. For CUB-200 (Wah et al., 2011), we initialize from ImageNet-pretrained weights and adopt a split learning rate: $1 \times 10^{-2}$ for the backbone and $1 \times 10^{-1}$ for the heads. The distiller and adapter are trained with learning rate $5 \times 10^{-2}$ and weight decay $1 \times 10^{-4}$. For the main from-scratch experiments, we set $\lambda_{\text{bi}}$=5 and $\lambda_{\text{cyc}}$=1. After Stage I, we fine-tune the adapter for 30 epochs using SGD with an initial learning rate of $1 \times 10^{-2}$ and weight decay $5 \times 10^{-4}$.

All other hyperparameters follow AdaGauss verbatim. In particular, we keep its prototype storage and sampling settings unchanged; the small additional adapter overhead is quantified in Sec. 4.7. For completeness, we note that the public AdaGauss code reports TinyImageNet results averaged over splits formed from the *first* 100 classes, which is slightly misaligned with common balanced partitions. To enable an apples-to-apples comparison, our tables present the corrected numbers under the standard balanced partitioning.

**Evaluation metrics.** We report three standard measures: the last-task average accuracy $A_{\text{last}}$, the average incremental accuracy $A_{\text{inc}}$, defined as the running mean of $A_{\text{last}}$ over incremental steps, and the last-task average forgetting $F_{\text{last}}$. Dataset specifics, hyper-parameter schedules, and metric definitions are provided in the Appendix A.3.2.

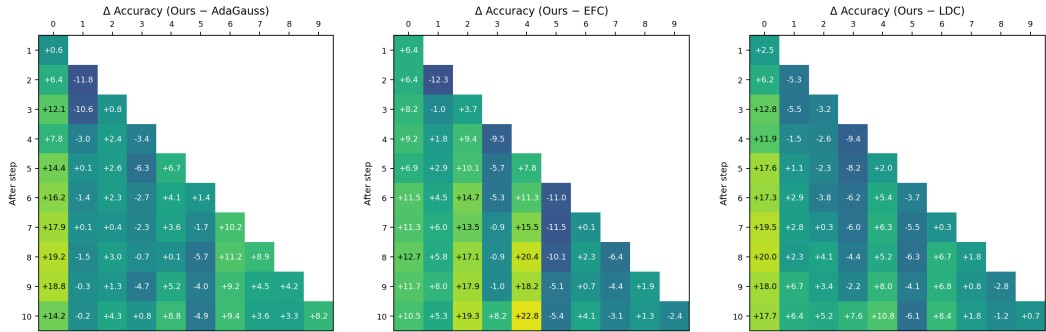

**Figure 3.** CIFAR-100 ($T$=10): Per-step, per-task accuracy gains ($\Delta$, percentage points) of **Ours** over Ada-Gauss, EFC, and LDC. Improvements concentrate on earlier tasks, indicating stronger retention and reduced forgetting.

## 4.1 MAIN RESULTS

Tables 1 and 2 report results on CIFAR-100, TinyImageNet, and ImageNet-100 with feature extractors trained from scratch, and on CUB-200 with ImageNet-pretrained initialization (mean±std over five runs). **CIFAR–100:** compared to AdaGauss, we gain **+3.8/+2.3** pp at $T$=10 and **+3.6/+2.1** pp at $T$=20. DPCR$^\ddagger$ is competitive, but our method still slightly leads on all CIFAR-100 settings (e.g., +0.4/+0.4 pp at $T$=10 and +1.7/+1.7 pp at $T$=20). **TinyImageNet:** improvements over Ada-Gauss are **+2.5/+3.3** pp at $T$=10 and **+2.7/+4.7** pp at $T$=20; the margins over the second-best (EFC) are +0.9/+1.2 pp ($T$=10) and +1.8/+2.1 pp ($T$=20). DPCR again trails our method, with gaps of about +1.1/+2.2 pp at $T$=10 and +4.6/+4.9 pp at $T$=20. **ImageNet–100:** vs. AdaGauss we obtain **+1.6/+1.8** pp at $T$=10 and **+1.2/+0.8** pp at $T$=20; at $T$=10 our $A_{\text{last}}$ is best (runner-up LDC$^\dagger$, +1.3 pp), while $A_{\text{inc}}$ is 2.6 pp below the best (LDC$^\dagger$). Under our protocol, rerunning public LDC code at $T$=10 yields $A_{\text{last}}$=41.7 ± 1.5% and $A_{\text{inc}}$=58.7 ± 1.7%. DPCR$^\ddagger$ is clearly weaker than AdaGauss and ours: at $T$=10 it trails our method by about 2.8/2.0 pp in $A_{\text{last}}/A_{\text{inc}}$ (and is already slightly below AdaGauss by 1.2/0.2 pp), while at $T$=20 the gap to ours further widens to 6.5/3.5 pp (with Ada-Gauss still ahead of DPCR by 5.3/2.7 pp). For $T$=20 we achieve the best $A_{\text{last}}$ and $A_{\text{inc}}$ (runner-up AdaGauss: +1.2/+0.8 pp). **CUB–200 (ImageNet pre-trained):** our performance is close to Ada-Gauss (vs. AdaGauss: +0.8/+0.6 pp at $T$=10, $-1.3/-1.1$ pp at $T$=20), while on the 20-split setting we trail EFC by 2.4/3.4 pp. DPCR does not report its CUB-200 hyperparameter configuration under our training protocol, so the corresponding entries are marked "–" in Table 2. With a pretrained backbone, practitioners typically adopt a very low backbone learning rate, which keeps cross-task feature drift small and thus limits the incremental gains of our method.

**Per-step advantage on CIFAR-100** ($T$=10). As shown in Figure 3, across three baselines, our method shows consistently positive accuracy gain throughout training, with the *largest gains on older tasks* (lower-right region in each heatmap). Against EFC, margins often exceed **+15–20** pp at mid/late steps; versus LDC, we sustain **+6–11** pp on most old tasks; and relative to AdaGauss we obtain **+5–10** pp improvements that persist to the

Table 4: CIFAR-100: Contributions of $\mathcal{L}_{\text{bi}}$ and $\mathcal{L}_{\text{cyc}}$.

| Components | | $T$=10 | | $T$=20 | |
|---|---|---|---|---|---|
| $\mathcal{L}_{\text{bi}}$ | $\mathcal{L}_{\text{cyc}}$ | $A_{\text{last}}(\%)$ | $A_{\text{inc}}(\%)$ | $A_{\text{last}}(\%)$ | $A_{\text{inc}}(\%)$ |
| × | × | 46.8±1.2 | 60.9±1.0 | 37.9±1.0 | 54.4±0.8 |
| ✓ | × | 49.4±1.0 | 62.1±1.1 | 40.2±1.1 | 55.8±1.0 |
| × | ✓ | 47.8±1.1 | 61.8±1.0 | 39.0±1.1 | 54.9±0.9 |
| ✓ | ✓ | **50.6±0.9** | **63.2±1.3** | **41.5±1.1** | **56.5±1.3** |

final step. The concentration of positive $\Delta$ on early tasks indicates **significantly smaller forgetting**: accuracy on initial tasks decays far less under ours while recent tasks remain competitive, yielding a superior plasticity–stability trade-off.

## 4.2 ADVANCE IN FORGETTING

As shown in Table 3, across the three **balanced, training-from-scratch** datasets, our method achieves the **lowest forgetting**. On **CUB-200**, however, most methods fine-tune from a **pretrained backbone**, so the gaps in forgetting are much smaller than in the from-scratch regime.

Table 5: Adapter Strategy vs. Architecture: Ablation on CIFAR-100

(a) Direct prototype projection vs. projection with post-training adapter fine-tuning. Arrows indicate the preferred direction.

| Ablation | $T{=}10$ | | $T{=}20$ | |
|---|---|---|---|---|
| | $A_{\text{last}} \uparrow$ | $F_{\text{last}} \downarrow$ | $A_{\text{last}} \uparrow$ | $F_{\text{last}} \downarrow$ |
| Direct projection | 49.9 | 15.2 | 38.9 | 17.6 |
| + fine tuning (vs. Direct) | +0.7 | -1.7 | +2.6 | -1.0 |

(b) Adapter/Distiller Architectures: MLP shows absolute scores, others report $\Delta$ vs. MLP

| Ablation | $T{=}10$ | | $T{=}20$ | |
|---|---|---|---|---|
| | $A_{\text{last}} \uparrow$ | $F_{\text{last}} \downarrow$ | $A_{\text{last}} \uparrow$ | $F_{\text{last}} \downarrow$ |
| MLP | 50.6 | 13.5 | 41.5 | 16.6 |
| Linear | -3.5 | +1.2 | -4.2 | +1.3 |
| CrossAttention | -2.7 | -0.7 | -6.0 | -1.0 |
| MoE | -3.7 | -3.6 | -2.8 | -4.9 |

### 4.3 EFFECT OF $\mathcal{L}_{\text{BI}}$ AND $\mathcal{L}_{\text{CYC}}$

Notably, our approach delivers **especially strong preservation of prior knowledge** when training from scratch.

As summarized in Table 4, on CIFAR-100 enabling either loss improves both $A_{\text{last}}$ and $A_{\text{inc}}$ over the AdaGauss baseline, and enabling both yields the best results across the 10- and 20-task splits. This pattern matches the roles established in Sec. 3: $\mathcal{L}_{\text{bi}}$ (Eq. 6) reduces the new$\leftrightarrow$old feature-transport errors that bound prototype mean/covariance mismatch, while $\mathcal{L}_{\text{cyc}}$ (Eq. 7) contracts the spectrum of $AD$ toward 1, mitigating rank loss and promoting near-isometric transport. Used together, they simultaneously lower transport error and preserve geometry, explaining consistent gains in $A_{\text{last}}$ and $A_{\text{inc}}$. Empirical diagnostics corroborate this: Figs. 4 and 5 (CIFAR-100, $T{=}10$) show lower symmetric KL between transported and ground-truth class Gaussians and singular-value spectra of $AD$ that are tighter and more concentrated at 1 than AdaGauss, indicating better distributional transport and more stable decision boundaries.

### 4.4 ABLATION: DIRECT PROJECTION VS. POST-TRAINING FINE-TUNING.

The adapter learned via bidirectional cycle consistency can be used *as is* to map old-class prototypes into the new space. We compare this "Direct projection" with an additional *post-training* fine-tuning of the adapter. On CIFAR-100, direct projection achieves $A_{\text{last}}{=}49.9$ and $F_{\text{last}}{=}15.2$ at 10-task split, and $A_{\text{last}}{=}38.9$ and $F_{\text{last}}{=}17.6$ at 20-task split. Fine-tuning yields consistent gains: $+0.7$ points in $A_{\text{last}}$ and $-1.7$ in $F_{\text{last}}$ at 10-task split, and a larger $+2.6$ / $-1.0$ at 20-task split. These results indicate that while the cycle-consistent adapter already provides a strong zero-shot projection, a brief post-training adjustment further aligns prototypes to the new feature geometry—an effect that becomes more pronounced as the task sequence lengthens.

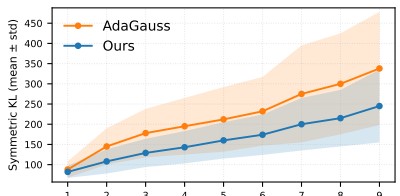

**Figure 4. Task-0 stability via SymKL ($\downarrow$).** On the fixed task-0 data, we compare Gaussian fits from models after $t{=}1\ldots9$ to the task-0 reference using symmetric KL (Eqs. 30–31); mean±std over classes. Our method maintains a smaller divergence—i.e., a closer match to the original distribution—than AdaGauss.

### 4.5 ABLATION: ADAPTER/DISTILLER ARCHITECTURE

Because our method learns bidirectional maps between old and new feature spaces, the adapter/distiller architecture directly affects performance. Beyond the linear or shallow MLP adapters common in prior work, we test lightweight but richer alternatives—cross-attention and sparse MoE—to probe whether conditional/nonlinear mappings better track representation drift. Table 5b reports CIFAR-100 results for the 10- and 20-task splits. Across both splits, multilayer adapters consistently outperform a single linear map: relative to an MLP baseline, the linear variant lowers $A_{\text{last}}$ by 3.5–4.2 points and increases $F_{\text{last}}$ by 1.2–1.3 points. Within the multilayer family, cross-attention favors stability, reducing forgetting ($\Delta F_{\text{last}} = -0.7$ to $-1.0$) at the expense of

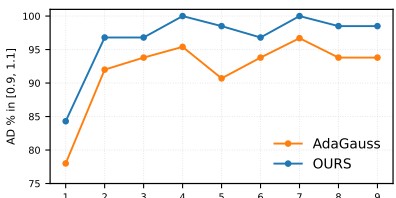

**Figure 5. Near-isometry on task-0 under continual updates.** AD-% in $[0.9, 1.1]$ for models after $t{=}1\ldots9$; our method consistently preserves geometry better than AdaGauss.

accuracy ($\Delta A_{\text{last}} = -2.7$ to $-6.0$), whereas sparse MoE delivers the largest forgetting gains ($-3.6$ to $-4.9$) with only moderate accuracy drops ($-2.8$ to $-3.7$). If new and old features differed by a single global affine transform, a linear adapter would suffice; the observed trade-offs instead point

to content-dependent, anisotropic drift, which conditional/nonlinear adapters model more faithfully. All variants share identical training schedules; a parameter-matched linear control is a natural follow-up to isolate capacity from architecture.

## 4.6 PROTOTYPE DRIFT FROM ORACLE MEANS ON CIFAR-100

To assess how well each method preserves old-class geometry, Fig. 6 reports prototype drift on CIFAR-100 with the 10-task split. After training Task 9, we freeze the backbone and, for every old class $c$, compute the maintained prototype $\widehat{\mu}_c$ and an *oracle* prototype $\mu_c^\star$ given by the empirical feature mean of all samples of class $c$ under the final backbone. The drift for class $c$ is defined as $\|\widehat{\mu}_c - \mu_c^\star\|_2$.

Panel 6a averages this drift over the ten classes of each source task, while Figure 6b plots the full per-class distribution over all 90 old classes. Our method yields both lower average drift and a tighter distribution at small values than AdaGauss and LDC, indicating less accumulated distortion of old-class prototypes.

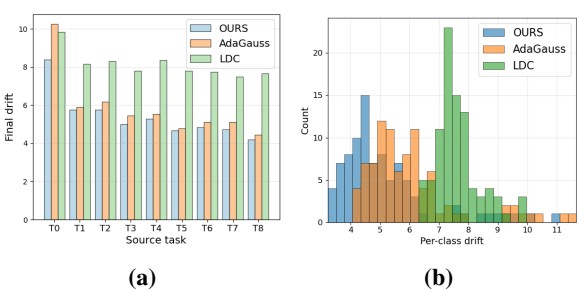

**(a)**        **(b)**

**Figure 6.** CIFAR-100 ($T{=}10$). Drift between maintained prototypes and oracle prototypes (empirical class means) after completing Task 9. For each of the 90 old classes (Tasks 0–8), we measure the $\ell_2$ distance in feature space between the maintained prototype and its oracle prototype. (a) Per-source-task average drift for the three methods. (b) Histogram of per-class drift over all old classes.

## 4.7 PARAMETER OVERHEAD IN THE 64-DIMENSIONAL SETTING

**Setup.** Following AdaGauss (Rypeść et al., 2024), all experiments use a ResNet-18 backbone followed by a $512{\to}64$ linear reduction and a two-layer MLP projector $D$ (new→old) in the $S{=}64$ space. Our bidirectional variant simply adds a second MLP $A$ (old→new) with the *same* architecture. Both $A$ and $D$ are MLPs $\mathbb{R}^S \to \mathbb{R}^{mS} \to \mathbb{R}^S$ with width multiplier $m{=}32$ (hidden size $mS{=}2048$).

**Parameter count.** A two-layer MLP with biases in this setting has

$$\#\mathrm{params}_{\mathrm{MLP}} = 2mS^2 + (m{+}1)S \quad \Rightarrow \quad \#\mathrm{params}_{\mathrm{MLP}} = 264{,}256$$

for $S{=}64$, $m{=}32$. Thus AdaGauss already uses one such projector $D$ ($\approx 0.26$M parameters), and our bidirectional version adds *one more* ($A$), for an extra

$$\Delta\#\mathrm{params} = 264{,}256$$

on top of the published AdaGauss model. Since a standard ResNet-18 backbone has about 11M parameters, the additional adapter increases the total parameter count by **only** $\approx 2.4\%$. We use this shared 64-dimensional configuration in all experiments.

## 5 CONCLUSIONS, LIMITATIONS, AND FUTURE WORK

**Conclusions.** We presented a bidirectional drift-compensation framework for exemplar-free class-incremental learning that jointly learns old to new and new to old projectors with stop-gradient gating and cycle consistency. Our analysis links least-squares projectors to CCA and shows how reducing alignment and cycle error stabilizes prototype margins. Experiments across standard EF-CIL benchmarks demonstrate strong forgetting reduction, especially in from-scratch settings, while maintaining competitive new-task accuracy.

**Limitations.** The current formulation assumes centered features, and second-order (Gaussian) prototype statistics; its theory is local to small alignment errors on the data support. The method may be sensitive to covariance estimation and hyperparameters in low-data regimes.

**Future work.** We plan to develop uncertainty-aware and class-imbalance–robust prototype transport, and derive non-asymptotic generalization/forgetting bounds beyond Gaussian assumptions. We also plan to integrate test-time adaptation and multi-modal backbones under strict memory budgets.

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

## A   APPENDIX

This appendix provides additional details, proofs, metrics, and supplementary results. It is organized as follows:

- **Method details.** §A.1.1 (transport view of drift compensation), §A.1.2 (Bayes classifier), §A.1.3 (pseudo-code).
- **Theory.** §A.2.1 (Proof of Theorem 1), §A.2.2 (Proof of Corollary 2).
- **Experimental protocol.** §A.3.1 (setup), §A.3.2 (accuracy metrics), §A.3.3 (distribution-similarity metrics), §A.3.4 (near-isometry metric).
- **Additional results and ablations.**   §A.4.1 (additional visualizations), §A.4.2 (distiller/adapter architecture ablations), §A.4.3 (hyperparameter sensitivity), §A.4.4 (choice of classifier).
- **Additional discussion and diagnostics.**   §A.5.1 (prototype-based EFCIL and Gaussian modeling in prior work), §A.5.2 (Gaussianity diagnostics), §A.5.3 (t-SNE snapshots), §A.5.4 (intuitive view of bidirectional cycle consistency), §A.5.5 (AdaGauss, full-covariance prototypes, and robustness).
- **Limitations and disclosure.** §A.6.1 (ImageNet-1K limitations), §A.6.2 (LLM usage disclosure).

### A.1   METHOD DETAILS

#### A.1.1   PROTOTYPE-DRIFT COMPENSATION: A TRANSPORT PERSPECTIVE

In the main paper, we adopt a vectorial notion of prototype drift. For each previously-seen class $c \in \mathcal{C}_{1:t-1}$, the backbone update from $f_{t-1}$ to $f_t$ induces the feature-mean displacement

$$\boldsymbol{\Delta}_c^t \; = \; \boldsymbol{\mu}_c^t - \boldsymbol{\mu}_c^{t-1}, \qquad \delta_c^t \; = \; \|\boldsymbol{\Delta}_c^t\|_2, \tag{15}$$

where $\boldsymbol{\mu}_c^t = \frac{1}{|\mathcal{D}_c|} \sum_{x \in \mathcal{D}_c} f_t(x)$ is the (unknown) class mean under the updated encoder $f_t$. Because EFCIL forbids storing past raw samples, $\boldsymbol{\mu}_c^t$ cannot be recomputed exactly, and cached prototypes $\boldsymbol{\mu}_c^{t-1}$ become stale once $f_t$ is deployed.

Most drift-compensation pipelines in prior work follow a two-stage pattern: Stage I constrains the backbone update using only $\mathcal{D}_t$,

$$\mathcal{L}_{\text{S1}} = \mathbb{E}_{(x,y) \in \mathcal{D}_t} \Big[ \text{CE}(g(f_t(x)), y) + \lambda \, D(\phi_{\text{new}}(x), \phi_{\text{old}}(x)) \Big], \tag{16}$$

with $\phi \in \{f(\cdot), \, g \circ f(\cdot)\}$ and $D$ a generic distillation/regularizer. Stage II then learns a forward adapter $A_t$ (with frozen $f_{t-1}, f_t$) by aligning paired features $(f_{t-1}(x), f_t(x))$ on $\mathcal{D}_t$, and transports old prototypes:

$$A_t \in \arg\min_A \mathbb{E}_{x \in \mathcal{D}_t} \|A(f_{t-1}(x)) - f_t(x)\|_2^2, \qquad \tilde{\boldsymbol{\mu}}_c^t \; = \; A_t(\boldsymbol{\mu}_c^{t-1}). \tag{17}$$

This transport view unifies existing drift-compensation recipes—each can be seen as instantiating either a global/class-wise translation $A_t(z) = z + \widehat{\boldsymbol{\Delta}}^t$ or a learned projector $A_t$ applied to cached prototypes.

**Transport-based summary of prior methods.** Below we cast representative EFCIL approaches as special cases of Eq. 17. For consistency, we denote the encoders by $f_{t-1}$ and $f_t$ (some works write $F_{t-1}, F_t$) and use $\mathcal{D}_t$ for the current-task data.

- **Semantic Drift Compensation (SDC)** (Yu et al., 2020). SDC estimates a *global* shift from new-task samples and uses it as a translation adapter:

$$\bar{\boldsymbol{\Delta}}^t = \frac{1}{|\mathcal{D}_t|} \sum_{x \in \mathcal{D}_t} (f_t(x) - f_{t-1}(x)), \qquad A_t(z) = z + \bar{\boldsymbol{\Delta}}^t, \quad \tilde{\boldsymbol{\mu}}_c^t = \boldsymbol{\mu}_c^{t-1} + \bar{\boldsymbol{\Delta}}^t.$$

- **Adversarial Drift Compensation (ADC)** (Goswami et al., 2024). For each old class $c$, ADC selects a current-sample $\hat{x}_c$ that is adversarially driven towards the vicinity of $\boldsymbol{\mu}_c^{t-1}$ (in the old space), and takes the resulting pairwise feature gap as a class-wise translation:

$$\widehat{\boldsymbol{\Delta}}_c^t = f_t(\hat{x}_c) - f_{t-1}(\hat{x}_c), \qquad A_t^{(c)}(z) = z + \widehat{\boldsymbol{\Delta}}_c^t, \quad \tilde{\boldsymbol{\mu}}_c^t = \boldsymbol{\mu}_c^{t-1} + \widehat{\boldsymbol{\Delta}}_c^t.$$

- **Learnable Drift Compensation (LDC)** (Gomez-Villa et al., 2024). LDC directly *learns* a projector as the adapter:

$$G_\theta \in \arg\min_G \mathbb{E}_{x \in \mathcal{D}_t} \|G(f_{t-1}(x)) - f_t(x)\|_2^2, \qquad A_t(z) = G_\theta(z), \quad \tilde{\boldsymbol{\mu}}_c^t = G_\theta(\boldsymbol{\mu}_c^{t-1}).$$

This captures non-linear, potentially class-dependent deformations.

- **EFC (EFM-weighted transport).** (Magistri et al., 2024) EFC computes a weighted average of per-sample shifts using a pseudo-metric induced by the Empirical Feature Matrix $E_{t-1}$ (estimated after task $t-1$). Let $\|v\|_E^2 := v^\top E\, v$. Each $x_i \in \mathcal{D}_t$ casts a vote for class $c$ with weight

$$w_{c,i} = \exp\left( - \frac{\| f_{t-1}(x_i) - \boldsymbol{\mu}_c^{t-1} \|_{E_{t-1}}^2}{2\sigma^2} \right),$$

yielding the class-wise transport

$$\widehat{\boldsymbol{\Delta}}_c^t = \frac{\sum_{x_i \in \mathcal{D}_t} w_{c,i}\, (f_t(x_i) - f_{t-1}(x_i))}{\sum_{x_i \in \mathcal{D}_t} w_{c,i}}, \qquad \tilde{\boldsymbol{\mu}}_c^t = \boldsymbol{\mu}_c^{t-1} + \widehat{\boldsymbol{\Delta}}_c^t.$$

- **AdaGauss.** Like LDC, AdaGauss first learns a forward projector $G_\theta$ by aligning paired features on $\mathcal{D}_t$:

$$G_\theta \in \arg\min_G \mathbb{E}_{x \in \mathcal{D}_t} \|G(f_{t-1}(x)) - f_t(x)\|_2^2, \qquad A_t(z) = G_\theta(z).$$

Unlike LDC—which directly transports old *means* via $\tilde{\boldsymbol{\mu}}_c^t = G_\theta(\boldsymbol{\mu}_c^{t-1})$—AdaGauss models each old class as a Gaussian and transports the *distribution* by Monte Carlo push-forward (see Alg. 1, Stage II):

$$u_m \sim \mathcal{N}(\boldsymbol{\mu}_c^{t-1}, \Sigma_c^{t-1}), \quad v_m = G_\theta(u_m) = A_t(u_m), \qquad m = 1, \ldots, M,$$

followed by re-estimation in the new space:

$$\tilde{\boldsymbol{\mu}}_c^t = \frac{1}{M} \sum_{m=1}^M v_m, \qquad \tilde{\Sigma}_c^t = \frac{1}{M-1} \sum_{m=1}^M (v_m - \tilde{\boldsymbol{\mu}}_c^t)(v_m - \tilde{\boldsymbol{\mu}}_c^t)^\top.$$

When $G_\theta$ (equivalently $A_t$) is affine, this reduces in closed form to pushing moments $(\tilde{\boldsymbol{\mu}}_c^t, \tilde{\Sigma}_c^t) = (A\boldsymbol{\mu}_c^{t-1} + b,\ A\Sigma_c^{t-1}A^\top)$.

### A.1.2 BAYES CLASSIFIER IN THE NEW FEATURE SPACE

Let $z = f_t(x) \in \mathbb{R}^d$ be the feature of an input $x$ at task $t$ and let each seen class $c \in \mathcal{C}_{1:t}$ be represented in the *new* space by a Gaussian prototype $\mathcal{N}(\mu_c, \Sigma_c)$ (means and covariances transported/estimated as in Sec. A.1.1). We define the Gaussian negative log-posterior score, up to constants independent of $c$, as

$$s_c(x) = (z - \mu_c)^\top \Sigma_c^{-1}(z - \mu_c) + \log \det \Sigma_c - 2\log \pi_c, \tag{18}$$

where $\pi_c$ is the class prior. In our implementation, we follow the common Mahalanobis-prototype variant and retain only the quadratic term, while the full Gaussian score above is used for the theoretical stability analysis; the same perturbation argument applies after dropping the prior and log-determinant terms. The *task-agnostic* prediction (TAg) is

$$\hat{y}_{\text{TAg}}(x) \;=\; \arg\min_{c \in \mathcal{C}_{1:t}} s_c(x). \tag{19}$$

When a task-aware (TAw) report is required, we restrict the argmin to the current task's label set $\mathcal{C}_t$:

$$\hat{y}_{\text{TAw}}(x) \;=\; \arg\min_{c \in \mathcal{C}_t} s_c(x). \tag{20}$$

### A.1.3 PSEUDO-CODE FOR OUR ALGORITHM

Algorithm 1 specifies the end-to-end procedure for each task $t$: it learns the current backbone $f_t$ under classification with bidirectional alignment and cycle consistency (via $A$ and $D$), and updates the class prototypes by transporting stored Gaussians into the current feature space for inference.

---

**Algorithm 1** Bidirectional Cycle Consistency (EFCIL)

---

**Inputs:** Task stream $\{\mathcal{D}_t\}_{t=1}^T$; old backbone $f_{t-1}$ (frozen); current backbone $f_t$ (learnable); classifier head $g$;
    adapter $A : \mathbb{R}^d \to \mathbb{R}^d$ (old→new); distiller $D : \mathbb{R}^d \to \mathbb{R}^d$ (new→old);
    hyperparameters $\lambda_{\text{bi}}, \lambda_{\text{cyc}}, \alpha$; learning rates $\eta, \eta_A, \eta_D$; batch size $B$;
    anti-collapse hyperparameters $\kappa, \beta, \varepsilon$; per-class sample count $M$ for distribution transport.
**Outputs:** Trained $f_t$, $A$, $D$ for each $t$; transported *means & covariances* for inference.

  **Initialization:** Copy $f_t \leftarrow f_{t-1}$; randomly initialize $A, D$; freeze $f_{t-1}$.
  **for** $t = 1, \ldots, T$ **do**
  *# Stage I: Joint training on current task $\mathcal{D}_t$*
    **while** *not converged* **do**
      Sample minibatch $\{(x,y)\}_{b=1}^B \sim \mathcal{D}_t$.
      $z_{\text{old}} \leftarrow f_{t-1}(x)$                                       ▷ no gradient
      $z_{\text{new}} \leftarrow f_t(x)$
      $\ell_{\text{new}} \leftarrow g(z_{\text{new}})$
      **Bidirectional alignment:**
      $\mathcal{L}_{\text{bi}} \leftarrow \|D(z_{\text{new}}) - z_{\text{old}}\|_2^2 + \|A(z_{\text{old}}) - z_{\text{new}}^{(\text{detach})}\|_2^2$
      **Cycle consistency:**
      $\mathcal{L}_{\text{cyc}} \leftarrow \|A(D(z_{\text{new}})) - z_{\text{new}}^{(\text{detach})}\|_2^2 + \|D(A(z_{\text{old}})) - z_{\text{old}}^{(\text{detach})}\|_2^2$
      **Classification:** $\mathcal{L}_{\text{CE}} \leftarrow \text{CE}(\ell_{\text{new}}, y)$
      **Robust anti-collapse on features:**

$$\Sigma \leftarrow \frac{1}{B-1}(z_{\text{new}} - \bar{z})^\top (z_{\text{new}} - \bar{z}); \quad \tilde{\Sigma} \leftarrow \frac{1}{2}(\Sigma + \Sigma^\top); \quad \hat{\Sigma} \leftarrow \tilde{\Sigma} + \kappa \frac{\text{tr}(\tilde{\Sigma})}{d} I + \varepsilon I$$

$$\mathcal{L}_{\text{ac}}^{\text{rob}} \leftarrow -\frac{1}{d} \sum_{i=1}^d \min(\text{chol}(\hat{\Sigma})_{ii}, \beta)$$

      **Total:** $\mathcal{L} \leftarrow \mathcal{L}_{\text{CE}} + \lambda_{\text{bi}}\mathcal{L}_{\text{bi}} + \lambda_{\text{cyc}}\mathcal{L}_{\text{cyc}} + \alpha\,\mathcal{L}_{\text{ac}}^{\text{rob}}$
    **end while**
  *# Stage II: Distribution transport via sampling + adapter fine-tuning (optional)*
    Freeze $f_{t-1}$, $f_t$, $D$; fine-tune $A$ on $\mathcal{D}_t$ with a small LR by minimizing $\|A(z_{\text{old}}) - z_{\text{new}}^{(\text{detach})}\|_2^2$.
    **for** each old class $c \in \mathcal{C}_{1:t-1}$ **do**
      Load stored stats $(\mu_c^{t-1}, \Sigma_c^{t-1})$.
      **Sample old features:** draw $U = \{u_m\}_{m=1}^M \sim \mathcal{N}(\mu_c^{t-1}, \Sigma_c^{t-1})$.
      **Push-forward to new space:** $V = \{v_m\}_{m=1}^M$ with $v_m \leftarrow A(u_m)$.
      **Re-estimate in new space:**
      $\tilde{\mu}_c^t \leftarrow \frac{1}{M} \sum_{m=1}^M v_m, \quad \tilde{\Sigma}_c^t \leftarrow \frac{1}{M-1} \sum_{m=1}^M (v_m - \tilde{\mu}_c^t)(v_m - \tilde{\mu}_c^t)^\top.$
    **end for**
    **Estimate new-class stats** under $f_t$ from $\mathcal{D}_t$: $(\mu_c^t, \Sigma_c^t)$ for all $c \in \mathcal{C}_t$.
    Build a new prototype collection using $\{(\tilde{\mu}_c^t, \tilde{\Sigma}_c^t)\}_{c \in \mathcal{C}_{1:t-1}}$ and $\{(\mu_c^t, \Sigma_c^t)\}_{c \in \mathcal{C}_t}$.
    **Store** $\{(\mu_c^t, \Sigma_c^t)\}_{c \in \mathcal{C}_{1:t}}$ for the next task.
  **end for**

---

## A.2 THEORY

### A.2.1 PROOF OF THEOREM 1

*Proof of Theorem 1 (Cycle contraction).* Let $M := \tilde{A}\tilde{D} - I$ and note that by definition of whitening, $\mathbb{E}[\tilde{z}_{\text{new}}\tilde{z}_{\text{new}}^\top] = I$ (features are taken to be centered; otherwise replace $z$ by its centered version). Then

$$\mathbb{E}\|M\,\tilde{z}_{\text{new}}\|_2^2 = \mathbb{E}[\tilde{z}_{\text{new}}^\top M^\top M \tilde{z}_{\text{new}}] = \text{Tr}(M^\top M\,\mathbb{E}[\tilde{z}_{\text{new}}\tilde{z}_{\text{new}}^\top]) = \text{Tr}(M^\top M) = \|M\|_F^2, \quad (21)$$

which yields the stated identity.

For the consequence, write the singular values of $\tilde{A}\tilde{D}$ as $\{\sigma_k\}_{k=1}^d$. Since $M = \tilde{A}\tilde{D} - I$, Weyl's inequality gives $\max_k|\sigma_k - 1| \le \|M\|_2 \le \|M\|_F$. Thus minimizing $\mathcal{L}_{\text{cyc}} = \mathbb{E}\|M\tilde{z}_{\text{new}}\|_2^2 = \|M\|_F^2$ forces $\|M\|_F \to 0$, hence $\sigma_k \to 1$ for all $k$. In particular, when the loss is small, all singular values of $\tilde{A}\tilde{D}$ lie in a tight neighborhood of 1, preventing rank/energy loss and preserving local geometry on the data support. $\qquad\square$

### A.2.2 PROOF OF COROLLARY 2

*Proof of Corollary 2 (Decision stability for classification).* Fix a class $c$ and abbreviate $\mu = \mu_c^t$, $\Sigma = \Sigma_c^t$, $\tilde{\mu} = \tilde{\mu}_c^t$, $\tilde{\Sigma} = \tilde{\Sigma}_c^t$, $\Delta\mu := \tilde{\mu} - \mu$, $\Delta\Sigma := \tilde{\Sigma} - \Sigma$. The Bayes log-score is $\ell_c(x) = \log \pi_c - \frac{1}{2}\log\det\Sigma - \frac{1}{2}(x-\mu)^\top\Sigma^{-1}(x-\mu)$. A first-order expansion in $(\Delta\mu, \Delta\Sigma)$ gives the perturbation

$$\tilde{\ell}_c(x) - \ell_c(x) = -\frac{1}{2}\,\text{Tr}(\Sigma^{-1}\Delta\Sigma) + \frac{1}{2}\,(x-\mu)^\top\Sigma^{-1}\Delta\Sigma\,\Sigma^{-1}(x-\mu) + \Delta\mu^\top\Sigma^{-1}(x-\mu) + R_c(x), \quad (22)$$

where $R_c(x) = O(\|\Delta\Sigma\|_2^2 + \|\Delta\mu\|_{\Sigma^{-1}}^2)$ by the identities $\log\det(\Sigma + \Delta\Sigma) = \log\det\Sigma + \text{Tr}(\Sigma^{-1}\Delta\Sigma) + O(\|\Delta\Sigma\|_2^2)$ and $(\Sigma + \Delta\Sigma)^{-1} = \Sigma^{-1} - \Sigma^{-1}\Delta\Sigma\,\Sigma^{-1} + O(\|\Delta\Sigma\|_2^2)$.

Taking absolute values and applying Cauchy–Schwarz and spectral norm bounds,

$$\begin{aligned}|\tilde{\ell}_c(x) - \ell_c(x)| \le\ & C_\Sigma^{(1)}\,\|\Delta\Sigma\|_2 + C_\Sigma^{(2)}\,\|\Delta\Sigma\|_2\,\|x-\mu\|_{\Sigma^{-1}}^2 \\ & + \|\Delta\mu\|_{\Sigma^{-1}}\,\|x-\mu\|_{\Sigma^{-1}} + O(\|\Delta\Sigma\|_2^2 + \|\Delta\mu\|_{\Sigma^{-1}}^2),\end{aligned} \quad (23)$$

for constants $C_\Sigma^{(1)}, C_\Sigma^{(2)}$ depending only on $\|\Sigma^{-1}\|_2$ (and dimension via standard inequalities). For a pair $(i,j)$, the log-odds perturbation satisfies by triangle inequality

$$|(\tilde{\ell}_i - \tilde{\ell}_j) - (\ell_i - \ell_j)| \le C_\mu(\|\Delta\mu_i\|_{(\Sigma_i^t)^{-1}} + \|\Delta\mu_j\|_{(\Sigma_j^t)^{-1}}) + C_\Sigma(\|\Delta\Sigma_i\|_2 + \|\Delta\Sigma_j\|_2) + O(\cdot), \quad (24)$$

where $C_\mu, C_\Sigma$ absorb bounded factors of $\|x - \mu_c^t\|_{(\Sigma_c^t)^{-1}}$ on the evaluation support. Now set $\delta_c := \|\tilde{\mu}_c^t - \mu_c^t\|_{(\Sigma_c^t)^{-1}}$ and invoke the transport-fidelity bounds used in the corollary,

$$\delta_c \lesssim \sqrt{\varepsilon_{\text{old}\to\text{new}}^2}, \qquad \|\tilde{\Sigma}^t - \Sigma^t\|_2 \lesssim C_1\sqrt{\varepsilon_{\text{old}\to\text{new}}^2} + C_2\,\varepsilon_{\text{cyc,new}}, \quad (25)$$

to obtain

$$|(\tilde{\ell}_i - \tilde{\ell}_j) - (\ell_i - \ell_j)| \lesssim C_\mu(\delta_i + \delta_j) + C_\Sigma\|\tilde{\Sigma}^t - \Sigma^t\|_2. \quad (26)$$

If the right-hand side is strictly smaller than the Bayes margin $m_{ij}(x) := |\ell_i(x) - \ell_j(x)|$, then the sign of the log-odds is unchanged and the Bayes decision between $i$ and $j$ at $x$ is preserved, as claimed. $\qquad\square$

## A.3 EXPERIMENTAL PROTOCOL

### A.3.1 EXPERIMENTAL SETUP

We utilize a workstation equipped with an NVIDIA RTX 6000 Ada GPU and a Xeon Gold 6448Y CPU to run all the experiments.

**Datasets.** We evaluate our method on four canonical continual-learning benchmarks CIFAR-100, TinyImageNet, ImageNet-100 and CUB-200. Each benchmark is instantiated with multiple class-incremental task splits so that each class/image is assigned to exactly one incremental task; no raw samples from previous tasks are revisited in later tasks; only the granularity of the partition changes. We use the official train/val (or test) partitions supplied with each dataset.

- **CIFAR-100** consists of 50,000 training and 10,000 test images of size $32 \times 32$ drawn from 100 classes.

- **Tiny-ImageNet** contains 100,000 training and 10,000 validation images at $64 \times 64$ resolution spanning 200 classes.

- **ImageNet-100** (also referred to as **ImageNet-Subset**) includes 130,000 training and 5,000 validation images preprocessed to $224 \times 224$ for 100 classes.

- **CUB-200** comprises 11,788 bird photographs—5,994 for training and 5,794 for testing—covering 200 fine-grained species. All images are center-cropped and resized to $224 \times 224$ to match ImageNet preprocessing.

**Testing.** All results are reported with a test batch size of 512 and no test-time augmentations.

### A.3.2 ACCURACY METRICS

We evaluate continual learning along three complementary axes: (i) aggregate predictive performance on seen tasks, (ii) distributional alignment between stored prototypes and the current test distribution, and (iii) near–isometry of the learned transport between old and new representations. This subsection formalizes the first axis.

We report the **last-task average accuracy** $A_{\text{last}}$, its running mean **average incremental accuracy** $A_{\text{inc}}$, and the **last-task average forgetting** $F_{\text{last}}$. Let $a_i^{(K)}$ denote accuracy on task $i$ after training up to task $K$, and let $|\mathcal{C}_i|$ be the number of classes introduced at step $i$. Then

$$A_{\text{last}} = \frac{\sum_{i=1}^{K} |\mathcal{C}_i| \, a_i^{(K)}}{\sum_{i=1}^{K} |\mathcal{C}_i|}, \qquad A_{\text{inc}} = \frac{1}{K} \sum_{j=1}^{K} A_{\text{last}}^{(j)}, \qquad F_{\text{last}} = \frac{\sum_{i=1}^{K} |\mathcal{C}_i| \, f_i^{(K)}}{\sum_{i=1}^{K} |\mathcal{C}_i|}, \qquad (27)$$

where $f_i^{(K)} = [\max_{1 \leq j \leq K} a_i^{(j)} - a_i^{(K)}]_+$ and $A_{\text{last}}^{(j)}$ is $A_{\text{last}}$ evaluated at step $j$.

### A.3.3 DISTRIBUTION-SIMILARITY EVALUATION METRICS

To study prototype drift, we compare stored Gaussian prototypes to test-time class statistics under the current backbone. Let $f_\theta(\cdot) \in \mathbb{R}^S$ denote the feature map, and for each class $c$ let $(\widehat{\mu}_c, \widehat{\Sigma}_c)$ be the stored prototype. Given a held-out set $\mathcal{D}_c^{\text{test}}$, we compute

$$\mu_c^\star = \frac{1}{|\mathcal{D}_c^{\text{test}}|} \sum_{x \in \mathcal{D}_c^{\text{test}}} f_\theta(x), \qquad \Sigma_c^\star = \text{Cov}(\{f_\theta(x) : x \in \mathcal{D}_c^{\text{test}}\}) \in \mathbb{R}^{S \times S}.$$

For numerical stability, all expressions involving covariances use Tikhonov regularization $\widetilde{\Sigma} := \Sigma + \varepsilon I$ with a small $\varepsilon > 0$.

We report three per-class discrepancies that emphasize complementary aspects of drift; lower values are better.

**(1) Prototype Mean Drift ($\mu$-L2).**

$$\mu\text{-L2}_c = \|\widehat{\mu}_c - \mu_c^\star\|_2. \qquad (28)$$

**(2) Covariance Drift (Frobenius).**

$$\Sigma\text{-F}_c = \|\widehat{\Sigma}_c - \Sigma_c^\star\|_F = \sqrt{\text{tr}\Big[(\widehat{\Sigma}_c - \Sigma_c^\star)^\top (\widehat{\Sigma}_c - \Sigma_c^\star)\Big]}. \qquad (29)$$

**(3) Symmetric KL Between Gaussians.**

$$D_{\mathrm{KL}}(\mathcal{N}(\mu_1, \Sigma_1) \,\|\, \mathcal{N}(\mu_2, \Sigma_2)) = \frac{1}{2}\Big[\mathrm{tr}(\Sigma_2^{-1}\Sigma_1) + (\mu_2 - \mu_1)^\top \Sigma_2^{-1}(\mu_2 - \mu_1) - S + \ln \frac{\det \Sigma_2}{\det \Sigma_1}\Big],$$
(30)

with $S$ the feature dimension and inverses/determinants taken on regularized covariances. We report the bi-directional form:

$$\mathrm{SymKL}_c \;=\; D_{\mathrm{KL}}\Big(\mathcal{N}(\widehat{\mu}_c, \widetilde{\Sigma}_c) \,\|\, \mathcal{N}(\mu_c^\star, \widetilde{\Sigma}_c^\star)\Big) + D_{\mathrm{KL}}\Big(\mathcal{N}(\mu_c^\star, \widetilde{\Sigma}_c^\star) \,\|\, \mathcal{N}(\widehat{\mu}_c, \widetilde{\Sigma}_c)\Big).$$
(31)

**Aggregation over a Class Set.** For any per-class statistic $m_c \in \{\mu\text{-L2}_c, \Sigma\text{-F}_c, \mathrm{SymKL}_c\}$ and class set $\mathcal{C}$ (e.g., a task slice), we report its mean and dispersion:

$$\overline{m} \;=\; \frac{1}{|\mathcal{C}|}\sum_{c \in \mathcal{C}} m_c, \qquad \mathrm{std}(m) \;=\; \sqrt{\frac{1}{|\mathcal{C}|}\sum_{c \in \mathcal{C}}(m_c - \overline{m})^2}.$$
(32)

### A.3.4 AD-% IN [0.9, 1.1]

Finally, to probe geometry preservation of the old↔new mapping, we measure the fraction of singular values of the composed map that lie in a tight unit band. Consistent with Sec. A.1.1, let $f_{t-1}$ and $f_t$ be the frozen previous and current encoders at task $t$, and let $A_t$ (old→new) and $D_t$ (new→old) be the learned maps. On a held-out split $\mathcal{V}_t$ restricted to the newly introduced classes $\mathcal{C}_t$, extract paired features

$$z_{\mathrm{old}} = f_{t-1}(x) \in \mathbb{R}^S, \qquad z_{\mathrm{new}} = f_t(x) \in \mathbb{R}^S, \qquad x \in \mathcal{V}_t, \; y(x) \in \mathcal{C}_t,$$

stack them as $Z_{\mathrm{old}}, Z_{\mathrm{new}} \in \mathbb{R}^{S \times N}$, and form least-squares surrogates:

$$\widehat{D}_t = (Z_{\mathrm{old}} Z_{\mathrm{new}}^\top)(Z_{\mathrm{new}} Z_{\mathrm{new}}^\top)^\dagger, \qquad \widehat{A}_t = (Z_{\mathrm{new}} Z_{\mathrm{old}}^\top)(Z_{\mathrm{old}} Z_{\mathrm{old}}^\top)^\dagger.$$

Let $\{\sigma_i\}_{i=1}^S = \sigma(\widehat{A}_t \widehat{D}_t)$ be the singular values. We report

$$\mathrm{AD\text{-}\%\ in\ }[0.9, 1.1] \;=\; 100 \times \frac{1}{S}\sum_{i=1}^S \mathbf{1}\Big\{ 0.9 \leq \sigma_i(\widehat{A}_t \widehat{D}_t) \leq 1.1 \Big\}.$$
(33)

(If $A_t$ or $D_t$ is a single linear layer, its weight can replace the corresponding surrogate.)

**Interpretation.** Higher AD-% in $[0.9, 1.1]$ indicates that $A_t D_t$ is closer to an isometry with less spectral shrinkage/expansion. This complements Sec. A.3.3: improved near–isometry typically coincides with lower symmetric KL, indicating better preservation of the class-conditional geometry across tasks.

## A.4 ADDITIONAL RESULTS AND ABLATIONS

### A.4.1 ADDITIONAL VISUALIZATIONS

Figure 7 tracks prototype drift on the fixed task-0 validation split over steps $t=1 \ldots 9$ (CIFAR-100, $T=10$), reporting the mean L2 shift of class centers ($\mu$-L2; Eq. 28) and the Frobenius change of covariances ($\Sigma$-Fro; Eq. 29) with mean±std across classes; smaller is better. Our method exhibits consistently lower center and covariance drift than AdaGauss, indicating closer alignment to the original task-0 distribution, i.e., reduced degradation of old-class statistics as $f_t$ evolves.

**Relation to the main findings.** These curves complement the diagnostics in Sec. A.3.3: we observe both lower symmetric KL between transported and ground-truth Gaussians and a higher fraction of singular values for $A_t D_t$ within $[0.9, 1.1]$ (near-isometry), each pointing to better distributional transport and geometry preservation under our bidirectional + cycle training.

### A.4.2 ADDITIONAL DETAILS ON DISTILLER/ADAPTER ARCHITECTURE ABLATIONS

**Setup and parity.** All adapter/distiller variants in Table 5b are trained under an identical data pipeline, optimization schedule, and loss configuration; only the *architectural family* of the adapter/distiller changes. Each map takes an $S$-dimensional feature and returns an $S$-dimensional output. Unless noted, dropout is disabled and LayerNorms use default $\epsilon$.

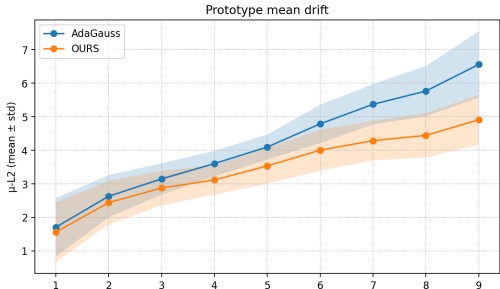
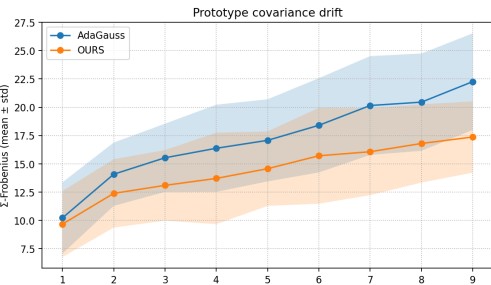

**Figure 7.** CIFAR-100 ($T$=10): **Prototype drift on task-0 under continual updates ($\downarrow$).** Using the fixed task-0 validation split, for each step $t$=1...9 we evaluate the model trained up to step $t$. Left: prototype mean drift $\mu$-L2 (Eq. 28); Right: covariance drift $\Sigma$-Frobenius (Eq. 29). Curves show mean$\pm$std over classes (Eq. 32); smaller is better. OURS exhibits consistently lower center and covariance drift than AdaGauss.

**Linear.** A single affine projection $W \in \mathbb{R}^{S \times S}$ without bias (i.e., $z \mapsto Wz$). This variant is parameter- and compute-light, and serves to illustrate the contribution of our objective under minimal capacity.

**MLP (default).** Unless stated otherwise, we instantiate the adapter/distiller as a *two-layer* MLP $\mathbb{R}^S \to \mathbb{R}^{mS} \to \mathbb{R}^S$ with GELU nonlinearity, no residual connection, and no dropout. We set the width multiplier to $m$=32 (hidden size $32S$), which matches the capacity used in our main experiments.

**Cross-Attention (XAttn).** We use a *single* cross-attention block with *pre-LayerNorm*, *8 heads*, and an FFN with *SwiGLU* and expansion $4\times$ (hidden size $4S$), followed by a linear projection back to $S$; dropout is disabled. Queries are produced from current (student) features and keys/values from frozen previous-task (teacher) features, following the standard encoder–decoder attention pattern (Vaswani et al., 2017).

**Mixture-of-Experts (MoE).** We optionally replace the projection MLP with a *sparse MoE* (Switch-style) comprising *4 experts*. A lightweight router (LayerNorm + linear) performs *top-1* routing per sample; the selected expert is a SwiGLU FFN with expansion $4\times$ (hidden size $4S$) and a linear projection back to $S$; dropout is disabled.

**Interpretation and scope.** Table 5b compares representative lightweight instantiations of Linear/MLP/XAttn/MoE under a common training protocol. Since parameter counts and FLOPs naturally co-vary across families, the absolute margins are best read as robustness across common, small-footprint configurations rather than as a strict capacity-matched ranking.

**Symbolic capacity accounting (per map).** Let $S$ denote the feature dimension and $mS$ the MLP hidden size. Ignoring biases, LayerNorm, and constants:

$$
\begin{aligned}
\text{Linear:} \quad & \Theta(S^2) \\
\text{2-layer MLP:} \quad & \Theta(2m\,S^2) \quad (S \to mS \to S; \text{ default } m\text{=}32) \\
\text{1-block XAttn:} \quad & \underbrace{4S^2}_{\text{Q/K/V/O}} + \underbrace{8S^2}_{\text{FFN (4}\times\text{)}} \approx 12S^2 \\
\text{Sparse MoE (4 experts, top-1):} \quad & \underbrace{O(S^2)}_{\text{router}} + \underbrace{8S^2}_{\text{active expert per sample}}
\end{aligned}
$$

### A.4.3 PARAMETER SENSITIVITY AND CHOICE OF DEFAULT HYPERPARAMETERS

In the main experiments we did not perform an extensive grid search. Instead, we chose the scales of the bidirectional and cycle-consistency losses based on their rough magnitude: $\lambda_{\mathrm{bi}}$=5 and $\lambda_{\mathrm{cyc}}$=1 were selected so that the additional terms had a similar order of contribution as the task loss and KD

Table 6: CIFAR-100: Sensitivity of $\lambda_{\text{bi}}$, $\lambda_{\text{cyc}}$, and $\alpha$.

| $\lambda_{\text{bi}}$ | $\lambda_{\text{cyc}}$ | $\alpha$ | $A_{\text{last}}(\%)$ | $A_{\text{inc}}(\%)$ | $A_{\text{last}}(\%)$ | $A_{\text{inc}}(\%)$ |
|---|---|---|---|---|---|---|
| | **Settings** | | | $T{=}10$ | | $T{=}20$ |
| 5 | 1 | 1 | 50.6 | 63.2 | 41.5 | 56.5 |
| 0 | 1 | 1 | 47.8 | 61.8 | 39.0 | 54.9 |
| 5 | 0 | 1 | 49.4 | 62.1 | 40.2 | 55.8 |
| 0 | 0 | 1 | 46.8 | 60.9 | 37.9 | 54.4 |
| 5 | 1 | 0 | 49.7 | 62.3 | 39.2 | 55.2 |
| 5 | 1 | 0.5 | 51.0 | 63.4 | 42.4 | 56.1 |
| 5 | 1 | 2 | 48.7 | 61.9 | 42.6 | 56.5 |
| 0.5 | 1 | 1 | 47.4 | 61.3 | 39.8 | 55.3 |
| 1 | 1 | 1 | 51.3 | **63.7** | 40.0 | 56.4 |
| 10 | 1 | 1 | 47.2 | 59.9 | 38.8 | 53.8 |
| 5 | 0.5 | 1 | 50.4 | 63.1 | 40.8 | 55.7 |
| 5 | 2 | 1 | **51.9** | 63.5 | **42.9** | **57.0** |

loss. For the anti-collapse loss we inherit the default scaling factor $\alpha{=}1$ from AdaGauss; our robust formulation preserves the role and typical scale of this regularizer.

The parameter-sensitivity study in Table 6 varies $\lambda_{\text{bi}} \in \{0, 0.5, 1, 5, 10\}$, $\lambda_{\text{cyc}} \in \{0, 0.5, 1, 2\}$, and $\alpha \in \{0, 0.5, 1, 2\}$. Alternative configurations can slightly outperform our default choice on CIFAR-100. Nevertheless, for all main results we retain the original defaults to avoid the impression that gains are driven by aggressive hyperparameter tuning. The bidirectional and cycle-consistency terms improve over the one-directional baseline in most non-degenerate settings, while overly large weights can hurt performance, as expected for strong auxiliary regularization.

### A.4.4 CHOICE OF CLASSIFIER

Table 7: Linear classifier vs. Bayesian classifier on CIFAR-100.

| Classifier | $T{=}10$ | | $T{=}20$ | |
|---|---|---|---|---|
| | $A_{\text{last}} \uparrow$ | $A_{\text{inc}} \uparrow$ | $A_{\text{last}} \uparrow$ | $A_{\text{inc}} \uparrow$ |
| Bayesian | 50.6 | 63.2 | 41.5 | 56.5 |
| Linear (sampling) | 51.1 | 63.7 | 40.8 | 55.7 |

Following common terminology in Gaussian-prototype EFCIL, we refer to our inference rule as the *Bayesian* classifier, although the implementation follows the Mahalanobis-prototype variant described in Sec. A.1.2: it retains the quadratic term of the Gaussian score and predicts by distance to stored Gaussian prototypes. We additionally evaluate a *linear* classifier trained from these Gaussians (Table 7). Following the public AdaGauss implementation, we construct a synthetic training set by sampling features from each class-wise Gaussian $\mathcal{N}(\mu_c, \Sigma_c)$ and optimize a single linear head over all seen classes with standard cross-entropy (denoted *Linear (sampling)*). In the EFC literature (Magistri et al., 2024; 2025), this procedure is often referred to as **Gaussian rebalancing**.

As shown in Table 7, the sampling-based linear head closely matches the Bayesian classifier: differences in $A_{\text{last}}$ and $A_{\text{inc}}$ remain below one percentage point for both $T{=}10$ and $T{=}20$. This indicates that our conclusions are not sensitive to the choice between the Bayesian/Mahalanobis-prototype inference rule and a Gaussian-sampling linear head.

### A.5 ADDITIONAL DISCUSSION AND DIAGNOSTICS

### A.5.1 PROTOTYPE-BASED EFCIL AND GAUSSIAN MODELING IN PRIOR WORK

Prototype-based strategies are a well-established line of work in exemplar-free class-incremental learning (EFCIL). Broadly, existing methods differ in how they represent class prototypes (means

vs. Gaussians) and how they use them (direct classification vs. pseudo-feature rehearsal vs. drift compensation).

**Mean prototypes with synthetic feature rehearsal.** PASS stores one feature mean per class and performs prototype rehearsal by injecting Gaussian noise around these means to synthesize pseudo-features, which are mixed with current-task data to train the classifier; a self-supervised rotation head is added to further stabilize the backbone. FeTrIL also stores class means, but produces old-class pseudo-features by a geometric translation of real features from the current task, $\hat{f} = f_{\text{new}} + \mu_{\text{old}} - \mu_{\text{new}}$, and uses these translated features together with new-class features to train a linear classifier.

**Explicit Gaussian prototypes and covariance-aware classification.** FeCAM estimates per-class means and covariances and performs Bayes/Mahalanobis classification in this Gaussian space. EFC represents each class as a Gaussian prototype $(\mu_c, \Sigma_c)$ and samples from these Gaussians to perform asymmetric prototype rehearsal (PR-ACE), while explicitly compensating prototype drift across tasks. AdaGauss likewise models each class as $\mathcal{N}(\mu_c, \Sigma_c)$ and introduces an anti-collapse regularizer based on the Cholesky factor of $\Sigma_c$ to prevent rank deficiency and feature collapse, together with covariance-adaptation mechanisms that update $(\mu, \Sigma)$ across tasks.

**Mean-only drift compensation.** LDC stores one mean prototype per class and learns a forward projector that maps old-space means into the new feature space after each task. ADC uses adversarially perturbed current-task inputs whose embeddings lie near old-class means and uses the resulting feature displacements to estimate how old means should move in the new space. Neither LDC nor ADC explicitly models full Gaussian structure.

**Our position.** Our work does not introduce Gaussian prototypes as a new concept; instead, we build on Gaussian-based EFCIL (FeCAM, EFC, AdaGauss) and mean-based drift compensation (LDC, ADC). Our novelty lies in how prototypes are **transported across tasks**: we introduce a bidirectional projector with cycle consistency that is trained online during backbone optimization, with theoretical guarantees linking the cycle loss to spectral contraction and classification stability.

### A.5.2 MEASURING THE ADHERENCE OF FEATURES TO GAUSSIAN ASSUMPTIONS

**On the suitability of multivariate normality tests.** Mardia's multivariate normality test is not well aligned with the geometry and scale of continual-learning vision features. It relies on third- and fourth-order moments (multivariate skewness and kurtosis), and its asymptotic calibration assumes moderate dimension and i.i.d. samples. In high-dimensional settings with complex dependence structures and large sample sizes—as in deep feature spaces of EFCIL benchmarks—this test can become overly restrictive and reject even when deviations are mild and do not affect downstream methods Ebner & Henze (2020); Chen & Xia (2023). Prior analyses also highlight sensitivity to sample size and dimensionality, motivating the use of graphical or geometric diagnostics.

Instead of relying on a global hypothesis test, we assess Gaussianity through a geometric, class-wise visualization in a low-dimensional embedding space. For a fixed subset of classes, we periodically extract their validation features across the training sequence, embed them with t-SNE, and overlay the corresponding fitted class-conditional Gaussians by plotting their one- and two-standard-deviation ellipses. This procedure directly reveals whether the learned representations form compact, approximately elliptical clusters that are stable over time.

### A.5.3 T-SNE SNAPSHOTS OF TASK-0 CLASSES ON CIFAR-100 (10 TASKS)

To better understand how feature distributions evolve over time, we conduct a t-SNE study on the *balanced* CIFAR-100 benchmark with $T=10$ equally sized tasks. We fix the ten classes introduced at task 0 and, after finishing tasks 0, 3, 6, and 9, extract their validation features and project them with t-SNE. For each snapshot in Figure 8, we fit a Gaussian to the features of each class and visualize its one- and two-standard-deviation regions with solid and dashed ellipses, respectively.

Across all stages, the per-class clusters remain roughly unimodal and are well covered by a single Gaussian, rather than fragmenting into multiple disjoint modes. This suggests that the dominant challenges in EFCIL are not due to a gross mismatch of the Gaussian prototype assumption, but rather due to drift of class statistics and reduced separability as the representation evolves.

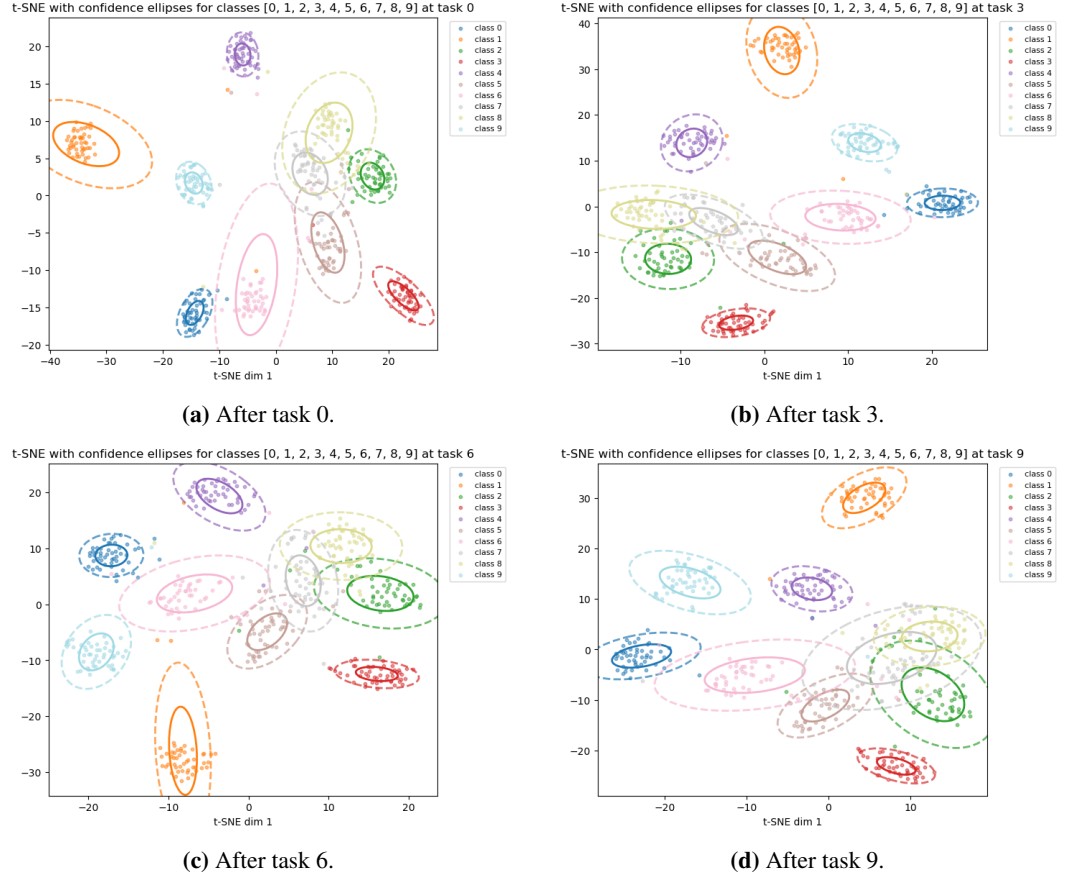

**(a)** After task 0.

**(b)** After task 3.

**(c)** After task 6.

**(d)** After task 9.

**Figure 8.** t-SNE of task-0 classes on CIFAR-100 with $T=10$. We project validation features of the same ten classes after training tasks 0, 3, 6, and 9. Solid and dashed ellipses mark the one- and two-standard-deviation regions of the fitted Gaussian for each class.

### A.5.4 Intuitive View of Bidirectional Cycle Consistency and Low-Drift Regimes

**From post-hoc adapters to in-task bidirectional alignment.** Most drift-compensation pipelines follow a two-stage pattern: during Stage I the new encoder $f_t$ is regularized toward $f_{t-1}$ (often via distillation), and only in Stage II is an adapter $A$ trained post hoc to map old features into the new space. Our goal is to make this duality explicit and move it inside Stage I: we jointly learn a distiller $D: z_{\text{new}} \to z_{\text{old}}$ and an adapter $A: z_{\text{old}} \to z_{\text{new}}$ while the backbone is still being optimized. Intuitively, $D$ regularizes $f_t$ toward the frozen teacher, while $A$ learns the forward transport used for prototype relocation under the evolving representation.

**Why this is not adversarial training.** Cycle consistency in Eq. 7 is a self-consistency constraint rather than an adversarial game: applying $D$ then $A$ (or $A$ then $D$) should approximately return the original feature on the data support. A key design choice is that both $L_{\text{bi}}$ and $L_{\text{cyc}}$ are implemented with stop-gradient targets: $\|A(z_{\text{old}}) - \text{stopgrad}(z_{\text{new}})\|^2$ updates $A$ only, so $A$ follows the evolving $f_t$ rather than dragging it; the cycle terms update $(A, D)$ but not $f_t$, stabilizing the maps without reducing backbone plasticity.

**An intuitive reading of Theorem 1 and Corollary 2.** Theorem 1 analyzes the cycle loss in a whitened feature space where each side has identity covariance; the expected cycle error is the squared Frobenius distance between the composed map $\tilde{A}\tilde{D}$ and the identity. Minimizing $L_{\text{cyc}}$ therefore encourages a near-isometry on the data support (singular values close to $1$), preventing rank/energy loss. Corollary 2 then links transport fidelity of means/covariances to bounded perturbations of Bayes scores; if the perturbation is smaller than the margin, old decisions are preserved.

**Why CUB-200 shows smaller gains.** On CUB-200 we fine-tune from an ImageNet-pretrained ResNet-18 with a very low backbone learning rate, so representation drift is modest. In this low-drift regime, the incremental benefit of aggressively regularizing and transporting features is naturally smaller; empirically the method behaves closer to AdaGauss.

### A.5.5 ON ADAGAUSS, FULL-COVARIANCE PROTOTYPES, AND THE NEED FOR ROBUSTNESS

**AdaGauss and full-covariance prototypes.** AdaGauss represents each class $c$ with a Gaussian prototype $\mathcal{N}(\mu_c, \Sigma_c)$ and uses these Gaussians both for classification (via a Bayes classifier) and for Gaussian sampling to train the prototype adapter. To make this feasible and numerically stable, AdaGauss introduces an anti-collapse loss that regularizes class-wise covariance matrices through a Cholesky factorization.

**Why dimensionality reduction is necessary.** The Cholesky-based anti-collapse term and Gaussian sampling require each $\Sigma_c$ to be symmetric positive-definite. In exemplar-free incremental settings, keeping the feature dimension at $512$ (ResNet-18 penultimate layer) often makes reliable full-rank covariance estimation difficult. AdaGauss therefore applies a learned linear reduction layer after the ResNet-18 backbone, mapping $512 \rightarrow 64$, improving the sample-to-dimension ratio and stabilizing covariance estimation.

**Why dimensionality reduction alone is still not sufficient.** Even after projecting to $S = 64$, the mini-batch covariance can be non-SPD or ill-conditioned in realistic EFCIL regimes (e.g., small $B$, correlated features, or imbalance), leading to Cholesky failures and unstable scales. We therefore adopt a robust anti-collapse variant: symmetrization, shrinkage, and jitter (and fallbacks when needed), aimed at guaranteeing numerical stability without changing the role of the regularizer.

**Implications for our method.** Following the public AdaGauss codebase, we retain the $512 \rightarrow 64$ projection. All Gaussian prototypes, anti-collapse losses, and transport maps are defined in the same $S = 64$ feature space, enabling direct comparisons and controlled ablations.

## A.6 LIMITATIONS AND DISCLOSURE

### A.6.1 LIMITATIONS OF IMAGENET-1K EXPERIMENTS

Under our current setup, scaling this protocol to the full 1K-class ImageNet dataset would require several weeks of continuous GPU time, making such an experiment unrealistic for the present study. Consequently, we restrict large-scale evaluation to ImageNet-100, whose class count still exposes the challenges of our setup while remaining computationally feasible.

### A.6.2 LLM USAGE DISCLOSURE

We used ChatGPT (OpenAI) as a writing copilot to critique and polish the prose (clarity, tone, and grammar). The model was not used to generate technical content, figures, or results, nor to design experiments or draw conclusions. The authors take full responsibility for all claims and the accuracy of the paper.

**We sincerely thank the readers for their time and careful reading of this paper.**

