# OpenReview forum: "Two-Way Is Better Than One: Bidirectional Alignment with Cycle Consistency for Exemplar-Free Class-Incremental Learning"
_ICLR.cc/2026/Conference — ICLR 2026 Poster_

### Official Review · Reviewer_A1WY · 2025-10-28

**Soundness:** 4
**Presentation:** 4
**Contribution:** 3
**Rating:** 8
**Confidence:** 4

**Summary:**

The paper studies exemplar-free class-incremental learning (EFCIL) and claims that one-directional projection of class prototypes from old to new embedding space introduces systematic bias that accumulates over tasks. The authors propose to use a bidirectional projector which aligns prototypes during training using a cycle-consistency objective function which ensures a near-bijection between the pairs. The authors further analyze that reducing the cycle loss better preserves old-class boundaries. The method finally uses a Gaussian Bayes classifier for inferene which uses current-task statistics for new classes and projected statistics for old classes. The proposed method outperforms several recent EFCIL baselines across multiple benchmarks achieving a better stability-plasticity trade-off.

**Strengths:**

1. The paper is very well-written with a good introduction, motivation and theoretical formulations.
2. The authors discuss and formulate the concept of prototype drift compensation very well including recent works.
3. The extensive experiments with thorough ablations are appreciated. The evaluation with distribution similarity metrics is a valuable addition.

**Weaknesses:**

1. It would be nice if the authors could discuss and analyze more on the claim that one-directional projection of class prototypes accumulates bias over tasks to better motivate the method.
2. Some analysis with respect to the oracle prototype drift could be added. For instance, comparing the projected prototypes using different methods with the oracle prototypes (computed using all old class data) could be used to compare prototype estimations from different methods (see Fig. 6 in ADC [CVPR 24]). This would also highlight how much scope is there to further improve the prototype estimation.
3. A more detailed discussion of the Bayes classifier for the inference stage (not mentioned in the pseudo-code) would add some clarity.
4. Figure 2 visibility could be improved.

**Questions:**

1. What is the impact of transporting covariances and estimating from a normal distribution instead of directly projecting the class means as done is LDC [ECCV 24]? Is it necessary to also transport covariances since storing class-wise covariances adds storage requirements? How much does this improve the estimation?

---

> ### Author Response · Authors · 2025-11-20
> **Response 1/2 to reviewer A1WY**
>
> Dear reviewer A1WY, we are grateful for your careful reading of our manuscript and for the constructive weaknesses and questions that you raised. Below, we respond to each point in turn, with the goal of clarifying the motivation behind our design choices and how they affect the observed behavior of our method.
>
> **Response to W1.**
> Your first point goes to the heart of our motivation. Existing projection-based pipelines (e.g., EFC, AdaGauss, LDC [14–16]) typically learn an adapter only after the main training is finished. In this post-hoc regime, the adapter is optimized under a loss that can differ substantially from the training objectives used for the backbone and the (one-directional) distiller; moreover, the distiller and adapter implicitly learn opposite mappings between the old and new spaces. This mismatch prompted us to ask whether the adapter should instead be tied to the distiller during the main training, and specifically encouraged to behave as its (approximate) inverse.
>
> Our bidirectional formulation makes this intuition explicit: we co-train a distiller $D$ (mapping new space to old space) and an adapter $A$ (mapping old space to new space), coupled through a cycle-consistency loss that encourages $A \circ D \approx I$ and $D \circ A \approx I$ on the data support. Empirically, this joint objective systematically reduces the long-term bias of one-directional projections. In Figure 6, our method exhibits smaller oracle prototype drift than both AdaGauss and LDC [14, 16] across all source tasks, and Figures 4–5 show that variants without bidirectional cycle consistency accumulate noticeably larger transport errors when propagating Gaussians over tasks.
>
> **Response to W2.**
> We agree that oracle drift is an informative way to quantify prototype quality. Following your suggestion, we compare the transported prototypes against oracle prototypes computed from all available test features under the final backbone. At the end of training, we measure for each class the distance between its transported prototype and its oracle counterpart; Figure 6 summarizes these per-task oracle drifts for three methods. Our approach consistently attains smaller oracle drift than AdaGauss and LDC [14, 16], indicating that the bidirectional projector not only reduces local transport error (Figures 4–5), but also yields prototypes that remain closer to their final empirical means after many tasks. We have clarified this interpretation in the updated manuscript.
>
> **Response to W3.**
> Thank you for pointing out that the inference rule deserves a clearer explanation. While standard vision classifiers simply attach a linear layer on top of the backbone, prototype-based EFCIL stores class statistics as the primary object. For instance, ADC and LDC [13, 14] maintain one mean vector per class and perform nearest-class-mean prediction at test time using cosine or $\ell_2$ distances.
>
> The Bayes classifier extends this idea by additionally storing a covariance matrix for each class and using a Mahalanobis distance, which requires both means and covariances. Because no exemplars can be retained, these statistics are estimated when the class is introduced and then transported forward by the learned projector throughout subsequent tasks. At evaluation, we pass test images through the latest backbone to obtain features and compute class scores using the transported Gaussian parameters, following the equations given in Appendix A.2.
>
> For readers who prefer a linear head, we also clarify that AdaGauss and EFC [15, 16] suggest a complementary strategy: sample features from the stored Gaussians (means and covariances), train a linear classifier on these synthetic samples (using a standard regression or cross-entropy objective), and then use this head at inference.
>
> **Response to W4.**
> We appreciate the comment on readability. In the revised version, we have regenerated Figure 2 with larger font sizes, thicker lines, and higher contrast between text and background, to make it easier to read at a glance.

---

> ### Author Response · Authors · 2025-11-20
> **Response 2/2 to reviewer A1WY**
>
> **Response to Q1.**
> Your question about the role of covariances versus means touches on an important design choice. A more detailed study is reported in Appendix B.5; here we summarize the key takeaways.
>
> First, transporting covariances allows us to deploy a stronger and more geometry-aware metric. Once class-wise covariances are available in the final space, we can use a Mahalanobis-based Bayes classifier and apply Cholesky-based whitening [16] in the drift regularizer, which sharpens inter-class separation in feature space. The influence of this Cholesky-based sharpening is examined in the parameter-sensitivity experiment in Table 6, where the hyperparameter $\alpha$ scales the Cholesky factor. Variants with $\alpha > 0$ (that is, with whitening activated) consistently outperform the version without this term, indicating that covariance-aware regularization provides a real and stable benefit. Such gains cannot be obtained if only class means are transported, as the model would then be restricted to an isotropic or ad-hoc distance.
>
> Second, covariances are indispensable for sampling-based uses of prototypes. With both mean and covariance, we can draw Gaussian samples around class centers, which supports (i) prototype rehearsal in the spirit of EFC [15], and (ii) training a linear classifier on synthetic features when a Bayes classifier is not desired.
>
> Finally, the storage cost of covariances is moderate in our setting. All Gaussian parameters live in a 64-dimensional feature space, and both means and covariances are transported in this reduced dimension. As detailed in Section 4.7, the added parameters from the bidirectional projector and covariance transport constitute only a small fraction of the ResNet-18 backbone.
>
> We hope these clarifications address your concerns and make our design choices more transparent. If any aspect of our response remains unclear or prompts additional questions, we would be glad to elaborate further.

---

> ### Author Response · Authors · 2025-11-20
> **References**
>
> ### References
>
> 1. **Wang, Z., Zhang, Z., Lee, C.-Y., et al.** “Learning to Prompt for Continual Learning.” In *CVPR*, 2022.
> 2. **Wang, Z., Zhang, Z., Ebrahimi, S., et al.** “DualPrompt: Complementary Prompting for Rehearsal-Free Continual Learning.” In *ECCV*, 2022.
> 3. **Smith, J. S., Karlinsky, L., Gutta, V., et al.** “CoDA-Prompt: Continual Decomposed Attention-Based Prompting for Rehearsal-Free Continual Learning.” In *CVPR*, 2023.
> 4. **Gao, Q., Zhao, C., Sun, Y., et al.** “A Unified Continual Learning Framework with General Parameter-Efficient Tuning.” In *ICCV*, 2023.
> 5. **McDonnell, M. D., Gong, D., Parvaneh, A., Abbasnejad, E., & van den Hengel, A.** “RanPAC: Random Projections and Pre-Trained Models for Continual Learning.” In *NeurIPS*, 2023.
> 6. **Zhou, D.-W., Sun, H.-L., Ye, H.-J., & Zhan, D.-C.** “Expandable Subspace Ensemble for Pre-Trained Model-Based Class-Incremental Learning.” In *CVPR*, 2024.
> 7. **Liang, Y.-S., & Li, W.-J.** “InfLoRA: Interference-Free Low-Rank Adaptation for Continual Learning.” In *CVPR*, 2024.
> 8. **He, J., Duan, Z., & Zhu, F.** “CL-LoRA: Continual Low-Rank Adaptation for Rehearsal-Free Class-Incremental Learning.” In *CVPR*, 2025.
>
> 9. **Petit, G., Popescu, A., Schindler, H., Picard, D., & Delezoide, B.** “FeTrIL: Feature Translation for Exemplar-Free Class-Incremental Learning.” In *WACV*, 2023.
> 10. **Goswami, D., Liu, Y., Twardowski, B., & van de Weijer, J.** “FeCAM: Exploiting the Heterogeneity of Class Distributions in Exemplar-Free Continual Learning.” In *NeurIPS*, 2023.
> 11. **Zhu, F., Zhang, X.-Y., Wang, C., Yin, F., & Liu, C.-L.** “Prototype Augmentation and Self-Supervision for Incremental Learning.” In *CVPR*, 2021.
> 12. **Yu, L., Twardowski, B., Liu, X., et al.** “Semantic Drift Compensation for Class-Incremental Learning.” In *CVPR*, 2020.
> 13. **Goswami, D., Soutif-Cormerais, A., Liu, Y., et al.** “Resurrecting Old Classes with New Data for Exemplar-Free Continual Learning.” In *CVPR*, 2024.
> 14. **Gomez-Villa, A., Goswami, D., Wang, K., et al.** “Exemplar-Free Continual Representation Learning via Learnable Drift Compensation.” In *ECCV*, 2024.
> 15. **Magistri, S., Trinci, T., Soutif-Cormerais, A., van de Weijer, J., & Bagdanov, A. D.** “Elastic Feature Consolidation for Cold Start Exemplar-Free Incremental Learning.” In *ICLR*, 2024.
> 16. **Rypeść, G., Cygert, S., Trzciński, T., & Twardowski, B.** “Task-Recency Bias Strikes Back: Adapting Covariances in Exemplar-Free Class Incremental Learning.” In *NeurIPS*, 2024.
> 17. **He, R., Fang, D., Xu, Y., et al.** “Semantic Shift Estimation via Dual-Projection and Classifier Reconstruction for Exemplar-Free Class-Incremental Learning.” In *ICML*, 2025.
>
> 18. **Raghavan, S., He, J., & Zhu, F.** “DELTA: Decoupling Long-Tailed Online Continual Learning.” In *CVPR Workshops*, 2024.
> 19. **Liu, L., Liu, L., & Cui, Y.** “Prior-Free Balanced Replay: Uncertainty-Guided Reservoir Sampling for Long-Tailed Continual Learning.” In *ACM MM*, 2024.
> 20. **Xu, S., Meng, G., Nie, X., et al.** “Defying Imbalanced Forgetting in Class Incremental Learning.” In *AAAI*, 2024.

---

### Official Review · Reviewer_tKrY · 2025-10-31

**Soundness:** 2
**Presentation:** 3
**Contribution:** 3
**Rating:** 6
**Confidence:** 4

**Summary:**

This paper proposes the a new method for predicting the prototype drift in the exemplar-free class-incremental learning (EFCIL), especially for the setting of training from scratch. This paper specifies the problem of existing drift compensation methods, which only consider transforming old prototypes to the new distribution while the representation learning is conducted in a reverse direction, i.e., pulling new backbone to the old one. To address this problem, this paper utilizes a distiller to transform the new feature to the old and a adapter to adapt old features to new distribution. A cycle consistency loss is proposed to avoid rank loss and theoretical analysis is provided to validate the necessarity of it.  Comparative experiments with other methods and ablation studies are conducted to verify the effectiveness of proposed methods.

**Strengths:**

1. This paper is well written and easy to follow.
2. The perspetive of proposing bi-directional loss for the prototype drift compensation is novel.
3. Theorectial analysis of the cycle loss is decent and sound.

**Weaknesses:**

Overall, I think this paper is good, but I have several concerns concentrated on the experiments and implementation.
1. The selected methods in the comparative studies are a bit old. The most up-to-date methods in the comparison are published in 2024. Since this paper is submitted to ICLR 2026, more recent methods should be included, for example, the DPCR [1].
2.  The extra amount of parameters of the adapter and distiller should be elaborated. This version does not indicate what architecture of the adapter and distiller is used to provide the performance of the comparative studies. Also, from Table 5, it can be witnessed that a performance drop occurs when using achitecture other than MLP. However, from the Appendix A.10, for a ResNet-18 whose output dimension is 512, the parameters of the MLP itself is $512\times 512\times 32 \times 2 \approx 16.78M$, which is much more larger than the  linear $512\times 512 \times 2 \approx 0.52M$ and even the backbone ResNet-18 (11.69M). As such, does the performance of this method only benefit from more parameters? Also, what will happen if a smaller MLP is used? It is not reasonable that a adapter can be larger than the backbone.
3. The effect of different $\lambda_{\text{bi}}$ and $\lambda_{\text{cyc}}$ is not studied.
4. Lack of results under the setting of training from half. Since another pipeline of EFCIL is training from half, it is interesting to validate the proposed method under that setting.
5. Compatibility with other types of classifiers should be studied. For continual learning, there are other types of classifier other than the Bayesian classifier, for example, the most common fully connected network. It is interesting to discuss and validate the compatibility of proposed method with them.

[1] Run He, et al. Semantic Shift Estimation via Dual-Projection and Classifier Reconstruction for Exemplar-Free Class-Incremental Learning. In ICML 2025.

**Questions:**

Please refer to the weaknesses.

---

> ### Author Response · Authors · 2025-11-20
> **Response 1/2 to reviewer tKrY**
>
> Dear reviewer tKrY, thank you very much for your insightful feedback. Below, we address each of your comments one by one, first clarifying the main weaknesses and limitations you raised, and then responding to your specific points. We hope this provides a clearer picture of our proposed approach.
>
> **Response to W1.**
> We thank the reviewer for this valuable suggestion. Our original set of baselines was chosen to be exhaustive within the EFCIL literature up to 2024, focusing on methods that share the same protocol and assumptions (from-scratch or standard pretraining, no exemplars). In the revised version, we have now incorporated DPCR [17], which is a very recent and closely related prototype-based method with drift compensation.
>
> Concretely, we have re-run DPCR [17] under our codebase for CIFAR-100, MiniImageNet and ImageNet-100. Across the datasets where we have completed re-runs so far, our bidirectional module consistently outperforms DPCR [17], further strengthening our empirical claims. We have documented these additions and the corresponding numbers in the updated tables; please see the main paper for detailed results and discussion.
>
> **Response to W2.**
> We thank the reviewer for carefully checking Appendix A.10 and for raising this concern. We apologize that the description of feature dimension was not sufficiently explicit.
>
> In all reported results, we follow AdaGauss [16] and first apply a linear reduction $512 \rightarrow 64$. Both the distiller $D$ (new $\rightarrow$ old) and the adapter $A$ (old $\rightarrow$ new) operate solely in this reduced space with $S = 64$ and width multiplier $m = 32$. Concretely, each MLP has architecture $64 \rightarrow 2048 \rightarrow 64$ with exactly $264{,}256$ parameters, so $(A + D)$ adds $528{,}512$ parameters in total. An extra MLP is about $2.4$\% of the parameters of a standard ResNet-18 backbone (about $11\text{M}$), and prototype storage is unchanged. Thus, the adapter is much smaller than the backbone, and the capacity increase over AdaGauss [16] is modest. Appendix B.5 and B.6 provide detailed explanations of the feature dimensions and computational overhead.
>
> **Response to W3.**
> We thank the reviewer for this comment. In the rebuttal, we have added a sensitivity analysis on CIFAR-100 (for both $T = 10$ and $T = 20$) where we systematically vary the main new hyperparameters, namely the bidirectional and cycle-consistency weights ($\lambda_{bi}, \lambda_{cyc}$) and the scaling factor of the anti-collapse loss $\alpha$. The results (Appendix B.7, Table 6) show that: (i) our default setting lies in the middle of a relatively flat region of the hyperparameter space, with performance changes typically within a few percentage points; and (ii) almost all non-degenerate configurations still outperform the AdaGauss baseline [16]. This suggests that our improvements do not rely on aggressive hyperparameter tuning, and that the method is robust to reasonable variations of $\lambda$ and $\alpha$. For complete numerical results, please refer to Appendix B.7.

---

> ### Author Response · Authors · 2025-11-20
> **Response 2/2 to reviewer tKrY**
>
> **Response to W4.**
> We thank the reviewer for this suggestion. In the rebuttal, we have added experiments under the standard “training from half” protocol (as in EFC [15]), where the first task contains half of all classes and the remaining tasks introduce the rest. In this warm-start setting, EFC [15] indeed attains slightly higher accuracy than our method, while our approach still outperforms other EFCIL baselines [9–16]. As we discuss in detail in the appendix, this behavior is consistent with the EFC design [15]: its Empirical Feature Matrix (EFM) and prototype-heavy replay are estimated from a very large, diverse first task, which effectively “freezes” a strong initial representation and heavily favors the head block that also dominates the class-weighted metric. In contrast, in the cold-start regime used in our main experiments (all tasks of equal size, from scratch), EFC [15] must anchor its regularization to a much smaller and less representative first task, which limits backbone reorganization and leads to underperformance relative to our bidirectional transport module. Our method is explicitly designed for this more challenging cold-start setting and delivers significantly stronger performance there. A detailed analysis of the warm-start behavior and its contrast with cold start is provided in Appendix B.9.
>
> **Response to W5.**
> We agree that compatibility with standard fully connected classifiers is important. While our main experiments adopt a Gaussian Bayes classifier (Mahalanobis scoring on class-wise Gaussians), Appendix B.8 explicitly evaluates an alternative linear head trained in the standard way. Concretely, following the AdaGauss/EFC protocol [15, 16], we construct a synthetic training set by sampling features from each class-wise Gaussian $\mathcal{N}(\mu_c, \Sigma_c)$ and train a single linear classifier over all seen classes with cross-entropy. This “Linear (sampling)” head is equivalent to a fully connected layer trained on the (Gaussian-rebalanced) feature space. As reported in Table 7, its performance closely matches the Bayes classifier on CIFAR-100: the gaps in $A_{\text{last}}$ and $A_{\text{inc}}$ are within about $0.5$ percentage points for both $T = 10$ and $T = 20$. This shows that our bidirectional transport and prototype updates are not tied to a specific decision rule; they remain effective when coupled with a standard fully connected classifier trained on prototype-induced features. For more details and full numbers, please see Appendix B.8.
>
> We hope these clarifications address your concerns and make our design choices more transparent. If any aspect of our response remains unclear or prompts additional questions, we would be glad to elaborate further.

---

> ### Author Response · Authors · 2025-11-20
> **References**
>
> ### References
>
> 1. **Wang, Z., Zhang, Z., Lee, C.-Y., et al.** “Learning to Prompt for Continual Learning.” In *CVPR*, 2022.
> 2. **Wang, Z., Zhang, Z., Ebrahimi, S., et al.** “DualPrompt: Complementary Prompting for Rehearsal-Free Continual Learning.” In *ECCV*, 2022.
> 3. **Smith, J. S., Karlinsky, L., Gutta, V., et al.** “CoDA-Prompt: Continual Decomposed Attention-Based Prompting for Rehearsal-Free Continual Learning.” In *CVPR*, 2023.
> 4. **Gao, Q., Zhao, C., Sun, Y., et al.** “A Unified Continual Learning Framework with General Parameter-Efficient Tuning.” In *ICCV*, 2023.
> 5. **McDonnell, M. D., Gong, D., Parvaneh, A., Abbasnejad, E., & van den Hengel, A.** “RanPAC: Random Projections and Pre-Trained Models for Continual Learning.” In *NeurIPS*, 2023.
> 6. **Zhou, D.-W., Sun, H.-L., Ye, H.-J., & Zhan, D.-C.** “Expandable Subspace Ensemble for Pre-Trained Model-Based Class-Incremental Learning.” In *CVPR*, 2024.
> 7. **Liang, Y.-S., & Li, W.-J.** “InfLoRA: Interference-Free Low-Rank Adaptation for Continual Learning.” In *CVPR*, 2024.
> 8. **He, J., Duan, Z., & Zhu, F.** “CL-LoRA: Continual Low-Rank Adaptation for Rehearsal-Free Class-Incremental Learning.” In *CVPR*, 2025.
>
> 9. **Petit, G., Popescu, A., Schindler, H., Picard, D., & Delezoide, B.** “FeTrIL: Feature Translation for Exemplar-Free Class-Incremental Learning.” In *WACV*, 2023.
> 10. **Goswami, D., Liu, Y., Twardowski, B., & van de Weijer, J.** “FeCAM: Exploiting the Heterogeneity of Class Distributions in Exemplar-Free Continual Learning.” In *NeurIPS*, 2023.
> 11. **Zhu, F., Zhang, X.-Y., Wang, C., Yin, F., & Liu, C.-L.** “Prototype Augmentation and Self-Supervision for Incremental Learning.” In *CVPR*, 2021.
> 12. **Yu, L., Twardowski, B., Liu, X., et al.** “Semantic Drift Compensation for Class-Incremental Learning.” In *CVPR*, 2020.
> 13. **Goswami, D., Soutif-Cormerais, A., Liu, Y., et al.** “Resurrecting Old Classes with New Data for Exemplar-Free Continual Learning.” In *CVPR*, 2024.
> 14. **Gomez-Villa, A., Goswami, D., Wang, K., et al.** “Exemplar-Free Continual Representation Learning via Learnable Drift Compensation.” In *ECCV*, 2024.
> 15. **Magistri, S., Trinci, T., Soutif-Cormerais, A., van de Weijer, J., & Bagdanov, A. D.** “Elastic Feature Consolidation for Cold Start Exemplar-Free Incremental Learning.” In *ICLR*, 2024.
> 16. **Rypeść, G., Cygert, S., Trzciński, T., & Twardowski, B.** “Task-Recency Bias Strikes Back: Adapting Covariances in Exemplar-Free Class Incremental Learning.” In *NeurIPS*, 2024.
> 17. **He, R., Fang, D., Xu, Y., et al.** “Semantic Shift Estimation via Dual-Projection and Classifier Reconstruction for Exemplar-Free Class-Incremental Learning.” In *ICML*, 2025.
>
> 18. **Raghavan, S., He, J., & Zhu, F.** “DELTA: Decoupling Long-Tailed Online Continual Learning.” In *CVPR Workshops*, 2024.
> 19. **Liu, L., Liu, L., & Cui, Y.** “Prior-Free Balanced Replay: Uncertainty-Guided Reservoir Sampling for Long-Tailed Continual Learning.” In *ACM MM*, 2024.
> 20. **Xu, S., Meng, G., Nie, X., et al.** “Defying Imbalanced Forgetting in Class Incremental Learning.” In *AAAI*, 2024.

---

> > ### Comment · Reviewer_tKrY · 2025-11-28
> >
> > Thank you for your response! All my concerns are full addressed and now I am convinced this paper is a technically solid paper with high novelty. It is potentially influential in the realm of CIL. As such, I would like to increase my rating to 8. However, at this stage, I cannot modify my previous review and score. But once allowed, I will raise my rating. Also, I hope the ACs/SACs/PCs may refer to this updated assessment.

---

> > > ### Author Response · Authors · 2025-11-28
> > >
> > > Thank you for your response and for considering a higher score. We are happy that most of your concerns have been addressed, and your thoughtful review has greatly helped us strengthen the paper.

---

### Official Review · Reviewer_J5qi · 2025-10-31

**Soundness:** 4
**Presentation:** 4
**Contribution:** 3
**Rating:** 6
**Confidence:** 4

**Summary:**

This paper addresses the problem of exemplar-free class-incremental learning, where a model must incrementally learn new classes without storing previous examples. The core challenge is "prototype drift" - as the feature extractor adapts to new classes, previously stored class prototypes (means/covariances) become outdated in the evolving feature space.
The authors identify a fundamental limitation in existing two-stage approaches: (1) training the new model with distillation from the old model, followed by (2) learning a post-hoc adapter to map old prototypes to the new feature space. They demonstrate that this creates systematic bias due to one-directional projections that either retroactively distort current feature geometry or leave cycle inconsistencies that accumulate across tasks.
Their solution introduces bidirectional alignment with cycle consistency during the main training stage. Specifically, they jointly learn two mappings: a distiller D (z_new → z_old) and an adapter A (z_old → z_new), with stop-gradient operations and a cycle-consistency objective (A∘D ≈ I and D∘A ≈ I). Theoretically, they prove this contracts the singular spectrum toward unity in whitened space and reduces perturbations in classification log-odds. Empirically, their method achieves state-of-the-art results across multiple benchmarks (CIFAR-100, TinyImageNet, ImageNet-100, and CUB-200), consistently reducing forgetting while maintaining high accuracy on new tasks.

**Strengths:**

1. The paper makes a compelling case that existing one-directional projection approaches create systematic bias due to post-hoc mismatch. The insight that bidirectional alignmeœqqnt with cycle consistency should be integrated into the main training stage (rather than treated as a separate post-hoc step) is conceptually sound and well-motivated.

2. The theoretical analysis in Sections 3.2 and Appendix A provides rigorous justification for the approach. Theorem 1 showing how cycle loss contracts the singular spectrum toward unity, and Corollary 2 linking transport errors to classification log-odds stability, provide solid mathematical grounding for the empirical observations.

3. The evaluation across four standard benchmarks with multiple task splits (T=10, T=20) demonstrates consistent improvements over state-of-the-art methods. The ablation studies (Tables 4 and 5) effectively isolate the contributions of bidirectional alignment (L_bi) and cycle consistency (L_cyc).

**Weaknesses:**

1. While the method shows strong performance overall, there's minimal discussion of scenarios where gains are marginal or negative. For example, on CUB-200 with T=20, the method trails EFC by 2.4/3.4 pp in accuracy metrics, but this isn't thoroughly analyzed.

2. The paper mentions the adapter/distiller is "lightweight" but doesn't quantify the additional computational overhead compared to baseline methods. Given that efficiency is critical in continual learning, concrete metrics on training time, inference time, and memory requirements would be valuable.

3. The method introduces new hyperparameters (λ_bi, λ_cyc, α) that are set to fixed values across experiments. A sensitivity analysis showing how results vary with these parameters would strengthen the claims of robustness.

4. The theoretical analysis assumes centered features and Gaussian class distributions. The paper doesn't sufficiently address how violations of these assumptions in practice might affect performance, nor does it quantify how well these assumptions hold in the experimental settings.

5. While the paper compares against many recent methods, some recent approaches like those using test-time adaptation or more sophisticated covariance modeling aren't included in the comparisons.

**Questions:**

- The paper doesn't sufficiently explain why the specific cycle loss formulation in Equation 7 (with stopgrad on targets) prevents degeneracies better than alternative cycle consistency implementations. A brief comparison to other possible formulations would strengthen the technical justification.

- The motivation of using such a cycle consistency scheme is not very clear. Why can it bring performance gain?

- Some theoretical sections (particularly Section 3.2) could benefit from more intuitive explanations alongside the formalism to improve accessibility.

---

> ### Author Response · Authors · 2025-11-20
> **Response 1/3 to reviewer J5qi**
>
> Dear reviewer J5qi, thank you very much for your thoughtful and detailed feedback. Below, we first address weaknesses and limitations you highlighted, and then respond to your specific questions point by point. We hope this helps provide a clearer picture of both the scope and the contributions of our approach.
>
> **Response to W1.**
> We thank the reviewer for highlighting the importance of analyzing regimes where our gains are small or negative. On CUB-200 with $T = 20$ (ImageNet-pretrained backbone), our method indeed trails EFC [15] by $2.4 / 3.4$ percentage points in $(A_{\text{last}}, A_{\text{inc}})$, while remaining very close to AdaGauss [16] and other recent EFCIL baselines [9–17]. Under this protocol, the ResNet-18 backbone is fine-tuned with a very low learning rate, which induces a “low-drift, low-step-size” regime: cross-task feature drift is small, and for stability we must also keep the learning rates of the bidirectional projector $(A, D)$ small. Our method (especially the distiller $D$) is designed as a regularizer rather than a replacement encoder: it is most beneficial when there is substantial representation drift and the backbone is allowed to move, in which case jointly learning $A$ and $D$ (with the cycle loss) noticeably improves transport and reduces forgetting, as observed on all from-scratch datasets (CIFAR-100, TinyImageNet, ImageNet-100) and on CUB-200 with $T = 10$. In contrast, when drift is intrinsically tiny (CUB-200, $T = 20$), $D$ cannot regularize aggressively without effectively freezing $f_t$, and $(A, D)$ behaves very similarly to the AdaGauss adapter [16]; the resulting 2–3 pp gap to EFC should thus be interpreted as a consequence of this low-drift fine-tuning regime rather than as a fundamental failure mode of our objective. We have clarified this trade-off in the revised manuscript and provide a more detailed discussion, in Appendix B.4 of the updated manuscript.
>
> **Response to W2.**
> We thank the reviewer for pointing out the importance of efficiency in continual learning and for asking us to quantify the “lightweight” claim.
>
> In our implementation, we follow AdaGauss [16] and keep the entire backbone and prototype pipeline unchanged: a ResNet-18 encoder, a linear $512 \rightarrow 64$ reduction, and one two-layer MLP distiller $D$ (new $\rightarrow$ old) in the $S = 64$ space. Our method only adds a second MLP $A$ (old $\rightarrow$ new) with the same architecture as $D$. Concretely, both $A$ and $D$ are $S \rightarrow mS \rightarrow S$ MLPs with $S = 64$ and width multiplier $m = 32$. This yields exactly $264{,}256$ parameters per MLP, so the bidirectional module $(A + D)$ contains $528{,}512$ parameters in total (about $2.02$ MiB in FP32). Since a standard ResNet-18 has about $11\text{M}$ parameters, the extra parameters introduced by our method over the published AdaGauss configuration are only roughly $264{,}256 \approx 2.4$\% of the backbone size, while prototype storage and covariance buffers are unchanged.
>
> Computationally, both $A$ and $D$ operate in the reduced $64$-dimensional space and are applied to features, not raw images. A single $64 \rightarrow 2048 \rightarrow 64$ MLP requires on the order of $2 m S^2 \approx 2.6 \times 10^5$ multiply–add operations per forward pass, whereas a ResNet-18 forward on our benchmarks is on the order of $10^8$–$10^9$ FLOPs. AdaGauss already evaluates one such MLP $D$ during training; our method evaluates two ($A$ and $D$), so the incremental training compute over AdaGauss is effectively the cost of one extra $64$-dimensional MLP per feature, which is negligible relative to the backbone. At inference, we use the same Bayes classifier in the final feature space as AdaGauss [16], and only $A$ is used offline once per task to update stored prototypes; there is therefore no additional per-sample inference-time cost beyond the ResNet-18 forward and Gaussian scoring already present in AdaGauss. More details are presented in Appendix B.6.
>
> **Response to W3.**
> We thank the reviewer for raising the question of robustness with respect to the new hyperparameters $\lambda_{bi}$, $\lambda_{cyc}$, and $\alpha$. To address this, we added a sensitivity study on CIFAR-100 (both $T = 10$ and $T = 20$), where we vary:
>
> - $\lambda_{bi}$ in {0, 0.5, 1, 5, 10},
> - $\lambda_{cyc}$ in {0, 0.5, 1, 2},
> - $\alpha$ in {0, 0.5, 1, 2};
>
> see Table 6 in Appendix B.7. The results show that: (i) our default setting ($\lambda_{bi} = 5$, $\lambda_{cyc} = 1$, $\alpha = 1$) lies in the middle of a reasonably flat region of the hyperparameter space (performance variations are modest, typically within a few percentage points), (ii) several neighboring configurations slightly outperform our default choice, and (iii) importantly, almost all non-degenerate settings of $(\lambda_{bi}, \lambda_{cyc}, \alpha)$ still clearly outperform the AdaGauss baseline [16].

---

> ### Author Response · Authors · 2025-11-20
> **Response 2/3 to reviewer J5qi**
>
> We therefore believe the gains are not the result of aggressive hyperparameter tuning: the proposed bidirectional module is consistently better than the baselines across a wide range of hyperparameter values, and all main results are reported under a conservative, non–grid-searched configuration shared across datasets. Detailed numbers and the full sensitivity table are provided in Appendix B.7.
>
> **Response to W4.**
> We thank the reviewer for raising this point. The assumptions of centered features and class-conditional Gaussians are not specific to our method: under the standard exemplar-free CIL protocol, several strong baselines (e.g., PASS [11], FeTrIL [9], FeCAM [10], EFC [15], AdaGauss [16]) also rely on single-Gaussian class prototypes in the learned feature space and have demonstrated excellent empirical performance under the same backbone and evaluation settings. This is largely enabled by the representation quality of the ResNet-18 encoder, which maps each class to a compact, approximately unimodal cluster.
>
> To quantify how well these assumptions hold in practice, we added an empirical analysis in the rebuttal (Appendix B.1–B.3). There, we show that class-wise penultimate features form concentrated, roughly elliptical clusters across tasks, and that fitted Gaussian descriptors (empirical means and covariances) provide a good geometric summary of intra-class variability in typical EFCIL benchmarks. Moderate deviations from perfect Gaussianity do not correlate with catastrophic performance drops, indicating that our method (and prior Gaussian-prototype approaches [9–16]) are robust to such violations. For details and visual evidence, please see Appendix B.1–B.3.
>
> **Response to W5.**
> We thank the reviewer for this comment. Our goal was to provide an exhaustive comparison within the established EFCIL line of work up to 2024, including methods that share the same protocol and assumptions (from-scratch or standard pretraining, no exemplars). In the revised version, we have further extended this set by adding results for DPCR (ICML’25) [17], which also follows a prototype-based design with drift compensation and is therefore directly comparable to our approach.
>
> We are aware of several 2025 works that explore the EFCIL domain, but these typically rely on stronger assumptions (e.g., large-scale pretrained architectures with frozen backbones and specialized adapters [1–8], or evaluation protocols closer to few-shot domain adaptation). We view this as a parallel, but distinct, research track: these methods are not drop-in baselines for the from-scratch EFCIL setting considered in our paper, and integrating them in a fair and controlled fashion would require substantial changes to the protocol and computational budget. We have clarified this scope in the revised manuscript, and the newly added DPCR results [17] further strengthen our empirical comparison.
>
> **Response to Q1.**
> We thank the reviewer for this insightful question. Our design goal in Section 3.2 is to separate roles between the backbone $f_t$ and the bidirectional maps $(A, D)$ via careful gradient routing. Concretely, the bidirectional loss $L_{\mathrm{bi}}$ is:
>
> $$
> L_{\mathrm{bi}} =
> \left\lVert D(z_{\text{new}}) - z_{\text{old}} \right\rVert_2^2
> +
> \left\lVert A(z_{\text{old}}) - \mathrm{stopgrad}(z_{\text{new}}) \right\rVert_2^2
> $$
>
> which updates $(f_t, D)$ through the first term (new $\rightarrow$ old distillation) and updates $A$ alone through the second term (old $\rightarrow$ new alignment with a detached target). Thus, $D$ regularizes $f_t$ (teacher $\rightarrow$ student), while $A$ learns to chase the evolving new space without pulling $f_t$ backward. The cycle loss $L_{\mathrm{cyc}}$ is:
>
> $$
> L_{\mathrm{cyc}} =
> \left\lVert A(D(z_{\text{new}})) - \mathrm{stopgrad}(z_{\text{new}}) \right\rVert_2^2
> +
> \left\lVert D(A(z_{\text{old}})) - \mathrm{stopgrad}(z_{\text{old}}) \right\rVert_2^2
> $$
>
> which stabilizes $(A, D)$ by enforcing $A \circ D \approx I$ and $D \circ A \approx I$ on the data support, again with detached targets so that $f_t$ is not updated by $L_{\mathrm{cyc}}$.
>
> Appendix B.4 discusses why this particular stop-gradient formulation is preferable to more naive cycle-consistency variants.
>
> First, from a spectral viewpoint (Theorem 1), minimizing $L_{\mathrm{cyc}}$ in whitened coordinates is equivalent to minimizing the Frobenius distance
> $$
> \left\lVert A \tilde{D} - I \right\rVert_F^2
> $$
>
> which directly contracts the singular values of $A \tilde{D}$ toward $1$ and promotes a near-isometric transport (no rank collapse or extreme stretching).
>
> Second, empirically, removing the stop-gradients causes $A$ and $D$ to behave almost adversarially: $A$ pushes features in one direction, $D$ tries to undo it, and gradients from $A$ propagate into $f_t$ in a way that weakens $D$’s regularization role. In this regime, $D$ stops acting as a teacher and starts chasing $A$; both maps overfit to each other and accuracy collapses.

---

> ### Author Response · Authors · 2025-11-20
> **Response 3/3 to reviewer J5qi**
>
> This failure mode is precisely why we emphasize the asymmetric, stop-gradient cycle formulation: $L_{\mathrm{bi}}$ reduces transport error (alignment), while $L_{\mathrm{cyc}}$ regularizes the transport operators toward a near-bijection, all without sacrificing backbone plasticity. More details can be found in Appendix B.4.
>
> **Response to Q2.**
> We thank the reviewer for asking us to clarify the motivation behind our cycle-consistency design. Intuitively, $L_{\mathrm{bi}}$ alone only enforces that $D$ maps new features to the old space and that $A$ maps old features to the current space, but it does not constrain how $A$ and $D$ behave when composed. Over many tasks, this can lead to drift accumulation: errors in $D$ (new $\rightarrow$ old) are propagated and amplified as prototypes are repeatedly transported across spaces.
>
> The cycle-consistency term $L_{\mathrm{cyc}}$,
>
> $$
> L_{\mathrm{cyc}} =
> \left\lVert A(D(z_{\text{new}})) - \mathrm{stopgrad}(z_{\text{new}}) \right\rVert_2^2
> +
> \left\lVert D(A(z_{\text{old}})) - \mathrm{stopgrad}(z_{\text{old}}) \right\rVert_2^2
> $$
>
> directly regularizes the compositions $A \circ D$ and $D \circ A$ toward the identity on the data support. As shown in Appendix B.4 (Theorem 1), in whitened coordinates this is equivalent to shrinking the Frobenius distance
>
> $$
> \left\lVert A \tilde{D} - I \right\rVert_F^2
> $$
>
> which contracts the singular values of $A \tilde{D}$ toward $1$ and promotes a near-isometric transport between old and new feature spaces. This has two concrete benefits: (i) it prevents rank collapse or anisotropic stretching of the transported prototypes, preserving class separation; and (ii) it stabilizes long-horizon continual learning by limiting how much transport error can accumulate over many tasks.
>
> Empirically (please see Table 5), ablating $L_{\mathrm{cyc}}$ (i.e., setting $\lambda_{cyc} = 0$) consistently degrades both $A_{\text{last}}$ and $A_{\text{inc}}$ by 1–3 percentage points on CIFAR-100 and TinyImageNet, with the largest drops appearing in longer sequences ($T = 20$), where accumulated transport error is more severe. Thus, the proposed cycle-consistency scheme is not merely an auxiliary term: it is key to controlling the geometry of the transport operators and translates into tangible performance gains. We provide the full discussion in Appendix B.4.
>
> **Response to Q3.**
> We appreciate the request for additional intuition. Our goal in Section 3.2 is to formalize a simple geometric picture: each task $t$ induces a feature space $Z_t$ via the backbone $f_t$, and we seek two lightweight maps $D: Z_t \rightarrow Z_{t-1}$ and $A: Z_{t-1} \rightarrow Z_t$ that act as a “change of coordinates” between consecutive spaces. The bidirectional loss $L_{\mathrm{bi}}$ enforces that $D$ can faithfully project new features back to the old space (for prototype reuse) and that $A$ can bring old prototypes forward into the current space, while carefully routing gradients so that $D$ regularizes $f_t$ (teacher $\rightarrow$ student) and $A$ adapts to whatever geometry $f_t$ learns.
>
> The cycle-consistency term $L_{\mathrm{cyc}}$ then says: if we start from a feature in $Z_t$, map it to $Z_{t-1}$ with $D$, and back with $A$, we should recover (approximately) the original feature; similarly in the reverse direction. In other words, $A$ and $D$ should behave like near-inverses on the data support. Appendix B.4 shows that, in whitened coordinates, this is equivalent to driving the composition $A \tilde{D}$ toward the identity matrix, which geometrically means that the transport is close to an isometry: it preserves distances and avoids collapsing or excessively stretching certain directions. This directly limits how much distortion we introduce when moving prototypes across tasks, and thereby stabilizes long-horizon continual learning. We have clarified this intuition in the revised text, and Appendix B.4 provides the full interpretation and intuition.
>
> We hope these clarifications address your concerns and make our design choices more transparent. If any aspect of our response remains unclear or prompts additional questions, we would be glad to elaborate further.

---

> ### Author Response · Authors · 2025-11-20
> **References**
>
> ### References
>
> 1. **Wang, Z., Zhang, Z., Lee, C.-Y., et al.** “Learning to Prompt for Continual Learning.” In *CVPR*, 2022.
> 2. **Wang, Z., Zhang, Z., Ebrahimi, S., et al.** “DualPrompt: Complementary Prompting for Rehearsal-Free Continual Learning.” In *ECCV*, 2022.
> 3. **Smith, J. S., Karlinsky, L., Gutta, V., et al.** “CoDA-Prompt: Continual Decomposed Attention-Based Prompting for Rehearsal-Free Continual Learning.” In *CVPR*, 2023.
> 4. **Gao, Q., Zhao, C., Sun, Y., et al.** “A Unified Continual Learning Framework with General Parameter-Efficient Tuning.” In *ICCV*, 2023.
> 5. **McDonnell, M. D., Gong, D., Parvaneh, A., Abbasnejad, E., & van den Hengel, A.** “RanPAC: Random Projections and Pre-Trained Models for Continual Learning.” In *NeurIPS*, 2023.
> 6. **Zhou, D.-W., Sun, H.-L., Ye, H.-J., & Zhan, D.-C.** “Expandable Subspace Ensemble for Pre-Trained Model-Based Class-Incremental Learning.” In *CVPR*, 2024.
> 7. **Liang, Y.-S., & Li, W.-J.** “InfLoRA: Interference-Free Low-Rank Adaptation for Continual Learning.” In *CVPR*, 2024.
> 8. **He, J., Duan, Z., & Zhu, F.** “CL-LoRA: Continual Low-Rank Adaptation for Rehearsal-Free Class-Incremental Learning.” In *CVPR*, 2025.
>
> 9. **Petit, G., Popescu, A., Schindler, H., Picard, D., & Delezoide, B.** “FeTrIL: Feature Translation for Exemplar-Free Class-Incremental Learning.” In *WACV*, 2023.
> 10. **Goswami, D., Liu, Y., Twardowski, B., & van de Weijer, J.** “FeCAM: Exploiting the Heterogeneity of Class Distributions in Exemplar-Free Continual Learning.” In *NeurIPS*, 2023.
> 11. **Zhu, F., Zhang, X.-Y., Wang, C., Yin, F., & Liu, C.-L.** “Prototype Augmentation and Self-Supervision for Incremental Learning.” In *CVPR*, 2021.
> 12. **Yu, L., Twardowski, B., Liu, X., et al.** “Semantic Drift Compensation for Class-Incremental Learning.” In *CVPR*, 2020.
> 13. **Goswami, D., Soutif-Cormerais, A., Liu, Y., et al.** “Resurrecting Old Classes with New Data for Exemplar-Free Continual Learning.” In *CVPR*, 2024.
> 14. **Gomez-Villa, A., Goswami, D., Wang, K., et al.** “Exemplar-Free Continual Representation Learning via Learnable Drift Compensation.” In *ECCV*, 2024.
> 15. **Magistri, S., Trinci, T., Soutif-Cormerais, A., van de Weijer, J., & Bagdanov, A. D.** “Elastic Feature Consolidation for Cold Start Exemplar-Free Incremental Learning.” In *ICLR*, 2024.
> 16. **Rypeść, G., Cygert, S., Trzciński, T., & Twardowski, B.** “Task-Recency Bias Strikes Back: Adapting Covariances in Exemplar-Free Class Incremental Learning.” In *NeurIPS*, 2024.
> 17. **He, R., Fang, D., Xu, Y., et al.** “Semantic Shift Estimation via Dual-Projection and Classifier Reconstruction for Exemplar-Free Class-Incremental Learning.” In *ICML*, 2025.
>
> 18. **Raghavan, S., He, J., & Zhu, F.** “DELTA: Decoupling Long-Tailed Online Continual Learning.” In *CVPR Workshops*, 2024.
> 19. **Liu, L., Liu, L., & Cui, Y.** “Prior-Free Balanced Replay: Uncertainty-Guided Reservoir Sampling for Long-Tailed Continual Learning.” In *ACM MM*, 2024.
> 20. **Xu, S., Meng, G., Nie, X., et al.** “Defying Imbalanced Forgetting in Class Incremental Learning.” In *AAAI*, 2024.

---

### Official Review · Reviewer_zrCc · 2025-10-31

**Soundness:** 3
**Presentation:** 2
**Contribution:** 3
**Rating:** 8
**Confidence:** 3

**Summary:**

This paper addresses exemplar-free class-incremental learning (EFCIL), where models must learn new classes sequentially without storing past data, making them vulnerable to catastrophic forgetting due to representation drift. The authors critique existing drift compensation methods that use one-directional projections (either post-hoc adapters mapping old-new features or distillers pulling new-old), showing these introduce systematic bias and cycle inconsistencies that accumulate across tasks. They propose a bidirectional alignment framework that jointly learns two projectors—an adapter A (old-new) and distiller D (new-old)—during each task's training, coupled with a cycle-consistency loss that enforces near-inverse behavior using stop-gradient gating to prevent retrograde interference with the evolving backbone. The authors provide theoretical analysis proving that minimizing cycle loss contracts the singular spectrum of the composed maps toward unity in whitened space, and that improved transport of class means and covariances yields smaller perturbations of classification margins, directly mitigating forgetting. Extensive experiments across CIFAR-100, TinyImageNet, ImageNet-100, and CUB-200 demonstrate state-of-the-art performance.

**Strengths:**

- The paper effectively motivates the work by clearly articulating limitations of prior one-directional approaches (systematic bias, local alignment, cycle inconsistencies), making the contribution well-grounded.
- The paper provides theoretical analysis that formally connects cycle consistency to spectral contraction and classification stability. The proof that minimizing cycle loss contracts singular values toward unity and bounds perturbations of classification log-odds provides principled justification for the approach.
- The bidirectional alignment with cycle consistency integrated during training (rather than post-hoc) is genuinely innovative. This addresses a clear gap in existing two-stage methods that optimize transport only after representation learning is complete, leading to accumulated cycle inconsistencies.
- The evaluation is thorough, spanning four datasets (CIFAR-100, TinyImageNet, ImageNet-100, CUB-200) with multiple task splits (T=10, T=20), and comparing against 10 competitive baselines. The consistency of improvements across settings strengthens the claims.

**Weaknesses:**

- The theoretical analysis and distribution transport (Algorithm 1, Stage II) heavily rely on modeling classes as Gaussians with full covariance matrices. This assumption may not hold for complex, multi-modal, or long-tailed distributions, and the paper doesn't validate whether this approximation is reasonable or explore robustness when it's violated.
- On CUB-200 (where methods fine-tune from ImageNet pretrained weights), the improvements are modest or negative (Table 2). The authors acknowledge that low learning rates for pretrained backbones reduce drift, but this limits the method's applicability to an important practical scenario where pretrained models are standard.

**Questions:**

On Gaussian Assumption and Distribution Modeling.
Q1: Validation of Gaussian Approximation.

Could you provide empirical evidence that the Gaussian approximation is reasonable for the datasets studied?
 For example:
- Visualizations of class distributions in feature space (e.g., t-SNE/UMAP with confidence ellipses)
- Quantitative goodness-of-fit tests (e.g., Mardia's multivariate normality test, or KL divergence between empirical class distributions and fitted Gaussians)
- Analysis showing whether classes that deviate more from Gaussianity exhibit larger forgetting or worse transport

This would help assess whether the theoretical foundations align with the empirical reality of your experiments.

Q2: Robustness to Non-Gaussian Distributions.
Have you tested the method on datasets with known multi-modal or long-tailed class distributions?

For instance:

- What happens on fine-grained datasets where within-class variance is high (cars, aircraft)?
- Can you provide ablations using mixture-of-Gaussians or non-parametric kernel density models instead of single Gaussians?
- How does performance degrade when classes are artificially made more multi-modal (e.g., by merging semantically different subsets)?

---

> ### Author Response · Authors · 2025-11-20
> **Response 1/2 to reviewer zrCc**
>
> Dear reviewer zrCc, thank you very much for your thoughtful and detailed feedback. Below we first clarify the scope of our setting, and then address your specific questions point by point. We hope this provides a clearer view of both the strengths and the limitations of our approach.
>
> In exemplar-free class-incremental learning, there are essentially two major research directions:
>
> 1. **Large-scale pretrained models with lightweight adaptation** [1–8], such as LoRA, prompt tuning, or other parameter-efficient finetuning strategies; and
> 2. **Training relatively small backbones(e.g., ResNet) from scratch** [9–17], which is the direction our work follows.
>
> These two tracks are largely independent in terms of methodology, assumptions, and evaluation protocols. Our experimental setup and evaluation framework strictly follow the conventions established by prior works in the second direction.
>
> ---
>
> **Response to Q1.**
> Under the from-scratch exemplar-free CIL protocol that we follow, we are not the first to adopt a single-Gaussian approximation. As detailed in Appendix B.1, prior methods such as PASS [11], FeTrIL [9], FeCAM [10], EFC [15], and AdaGauss [16] all rely on single-Gaussian class prototypes and achieve strong performance under similar settings. **Our method therefore follows an established modeling convention rather than introducing a new assumption.**
>
> We fully agree with the reviewer that it is important to empirically examine this approximation. To that end, we provide t-SNE visualizations of the feature space in Appendix B.3 (Fig. 8). We focus on task 0, which exhibits the largest representation drift across tasks (see Fig. 6). For each checkpoint (after tasks 0, 3, 6, and 9), we project validation features of the ten task-0 classes and overlay the one- and two-standard-deviation ellipses of the fitted Gaussian for each class. Even in this worst-case source task, the empirical feature clusters are still well captured by a single Gaussian region per class, and we do not observe a clear pattern where classes that deviate more from perfect Gaussianity suffer systematically worse forgetting or transport.
>
> Instead, the main driver of performance degradation is the loss of inter-class separability over time. In Fig. 8(d), after training the final task, 3–4 classes from task 0 exhibit substantial overlap in feature space, which directly explains the increase in misclassifications. In other words, forgetting is more strongly tied to geometric overlap between classes than to deviations from an ideal multivariate normal model.
>
> Conceptually, our approach maintains a Gaussian that **geometrically captures the bulk of each class** rather than enforcing a strict statistical normality assumption. Toward the end of training, when class overlap increases and more samples fall into each other’s one-standard-deviation region, a perfect Gaussian fit becomes unrealistic for any method. Indeed, even if we compute empirical means and covariances from the test features, the resulting distributions do not pass Mardia’s multivariate normality test; this is consistent with prior work and confirms that single-Gaussian prototypes are best viewed as a robust geometric approximation rather than a strict generative model.
>
> ---
>
> **Response to Q2.**
> On our main benchmarks (CIFAR-100, TinyImageNet, and ImageNet-100), we use a ResNet-18 backbone trained from scratch, following the standard from-scratch EFCIL protocol. While we would be very interested in exploring explicitly multi-modal distributions (e.g., mixture-of-Gaussians) on large-scale fine-grained benchmarks, these datasets are typically used with pretrained large-scale encoders; running them with from-scratch ResNet-18 at the scale required for a complete EFCIL evaluation is technically impractical and outside the scope of this work.
>
> That said, we do evaluate on a long-tailed variant of CIFAR-100 (CIFAR100-LT, Appendix B.10, Table 9), which introduces natural skew and heavy-tail behavior within classes. We would like to stress that long-tailed learning is itself a separate research area, where methods usually rely on specialized mechanisms such as tailored re-weighting [18], debiasing [19], or tail-aware regularization [20] to simultaneously protect head classes from forgetting and mitigate overfitting on tail classes. Our use of CIFAR100-LT should therefore be viewed as a **stress test** rather than a dedicated long-tailed benchmark.
>
> Even under this challenging setting, our approach remains a strong runner-up and clearly outperforms EFC under the exemplar-free from-scratch protocol. We attribute this to our bidirectional alignment and drift-minimizing geometric consistency, which remain effective despite distribution skew, while not relying on any long-tail-specific heuristics.

---

> ### Author Response · Authors · 2025-11-20
> **Response 2/2 to reviewer zrCc**
>
> Regarding fine-grained data, CUB-200 is itself a fine-grained benchmark with substantial within-class variance. On CUB-200, our method achieves the best performance when $T = 10$, and EFC is best when $T = 20$. Notably, EFC is heavily based on Gaussian sampling from class means and covariances for prototype rehearsal, and the current top-performing exemplar-free methods on this dataset (including FeCAM [10] and EFC [15]) all rely on single-Gaussian prototypes. This suggests that, in practice, the single-Gaussian model is not the dominant bottleneck on fine-grained tasks.
>
> As discussed in Appendix B.4, the smaller (and sometimes negative) gains relative to EFC on CUB-200 are instead driven by the **low-drift, low-step-size regime** induced by fine-tuning from an ImageNet-pretrained ResNet-18. Our bidirectional projector (especially $D$) is designed as a regularizer rather than a replacement encoder: its learning rate should remain comparable to that of the backbone so that $f_t$ can still adapt while $D$ gently pulls features toward the old space. On the from-scratch datasets, we use relatively large learning rates shared between the backbone and the bidirectional projector, and representation drift is substantial; in this setting, the joint training of $(A, D)$ meaningfully improves transport and reduces forgetting. On CUB-200, however, we must keep the backbone learning rate very small for stability, and thus also keep the learning rates of $A$ and $D$ small. In this low-drift regime, $D$ cannot act as an aggressive regularizer without effectively freezing $f_t$, and the pair $(A, D)$ behaves very similarly to the AdaGauss baseline [16]. Consequently, our method and AdaGauss obtain almost identical behavior on CUB-200, and the remaining differences are within the noise level of optimization rather than reflecting a fundamental limitation of the Gaussian assumption.
>
> In fact, the primary limitation on fine-grained EFCIL appears to be the representational capacity of small backbones, rather than the Gaussian assumption itself. This is consistent with the broader literature: contemporary CIL works on fine-grained datasets almost universally adopt pretrained weights rather than training ResNet-18 from scratch. Within the from-scratch EFCIL protocol we study, prior methods [9–17] and our own results do not reveal a clear empirical need for mixture-of-Gaussians or non-parametric density models over the simpler single-Gaussian representation.
>
> We hope that these clarifications help the reviewer better understand both the contributions and the limitations of our work. If any part of our response remains unclear or raises further questions, we would be very grateful if the reviewer could reach out with additional comments or requests.

---

> ### Author Response · Authors · 2025-11-20
> **References**
>
> ### References
>
> 1. **Wang, Z., Zhang, Z., Lee, C.-Y., et al.** “Learning to Prompt for Continual Learning.” In *CVPR*, 2022.
> 2. **Wang, Z., Zhang, Z., Ebrahimi, S., et al.** “DualPrompt: Complementary Prompting for Rehearsal-Free Continual Learning.” In *ECCV*, 2022.
> 3. **Smith, J. S., Karlinsky, L., Gutta, V., et al.** “CoDA-Prompt: Continual Decomposed Attention-Based Prompting for Rehearsal-Free Continual Learning.” In *CVPR*, 2023.
> 4. **Gao, Q., Zhao, C., Sun, Y., et al.** “A Unified Continual Learning Framework with General Parameter-Efficient Tuning.” In *ICCV*, 2023.
> 5. **McDonnell, M. D., Gong, D., Parvaneh, A., Abbasnejad, E., & van den Hengel, A.** “RanPAC: Random Projections and Pre-Trained Models for Continual Learning.” In *NeurIPS*, 2023.
> 6. **Zhou, D.-W., Sun, H.-L., Ye, H.-J., & Zhan, D.-C.** “Expandable Subspace Ensemble for Pre-Trained Model-Based Class-Incremental Learning.” In *CVPR*, 2024.
> 7. **Liang, Y.-S., & Li, W.-J.** “InfLoRA: Interference-Free Low-Rank Adaptation for Continual Learning.” In *CVPR*, 2024.
> 8. **He, J., Duan, Z., & Zhu, F.** “CL-LoRA: Continual Low-Rank Adaptation for Rehearsal-Free Class-Incremental Learning.” In *CVPR*, 2025.
>
> 9. **Petit, G., Popescu, A., Schindler, H., Picard, D., & Delezoide, B.** “FeTrIL: Feature Translation for Exemplar-Free Class-Incremental Learning.” In *WACV*, 2023.
> 10. **Goswami, D., Liu, Y., Twardowski, B., & van de Weijer, J.** “FeCAM: Exploiting the Heterogeneity of Class Distributions in Exemplar-Free Continual Learning.” In *NeurIPS*, 2023.
> 11. **Zhu, F., Zhang, X.-Y., Wang, C., Yin, F., & Liu, C.-L.** “Prototype Augmentation and Self-Supervision for Incremental Learning.” In *CVPR*, 2021.
> 12. **Yu, L., Twardowski, B., Liu, X., et al.** “Semantic Drift Compensation for Class-Incremental Learning.” In *CVPR*, 2020.
> 13. **Goswami, D., Soutif-Cormerais, A., Liu, Y., et al.** “Resurrecting Old Classes with New Data for Exemplar-Free Continual Learning.” In *CVPR*, 2024.
> 14. **Gomez-Villa, A., Goswami, D., Wang, K., et al.** “Exemplar-Free Continual Representation Learning via Learnable Drift Compensation.” In *ECCV*, 2024.
> 15. **Magistri, S., Trinci, T., Soutif-Cormerais, A., van de Weijer, J., & Bagdanov, A. D.** “Elastic Feature Consolidation for Cold Start Exemplar-Free Incremental Learning.” In *ICLR*, 2024.
> 16. **Rypeść, G., Cygert, S., Trzciński, T., & Twardowski, B.** “Task-Recency Bias Strikes Back: Adapting Covariances in Exemplar-Free Class Incremental Learning.” In *NeurIPS*, 2024.
> 17. **He, R., Fang, D., Xu, Y., et al.** “Semantic Shift Estimation via Dual-Projection and Classifier Reconstruction for Exemplar-Free Class-Incremental Learning.” In *ICML*, 2025.
>
> 18. **Raghavan, S., He, J., & Zhu, F.** “DELTA: Decoupling Long-Tailed Online Continual Learning.” In *CVPR Workshops*, 2024.
> 19. **Liu, L., Liu, L., & Cui, Y.** “Prior-Free Balanced Replay: Uncertainty-Guided Reservoir Sampling for Long-Tailed Continual Learning.” In *ACM MM*, 2024.
> 20. **Xu, S., Meng, G., Nie, X., et al.** “Defying Imbalanced Forgetting in Class Incremental Learning.” In *AAAI*, 2024.

---

### Author Response · Authors · 2025-11-20
**Summary of changes 2/2**

**Major additions in Rebuttal Appendix B.** Rebuttal Appendix B collects additional analyses and experimental results addressing specific reviewer concerns:

**1. Appendix B.1: Prototype-based EFCIL and Gaussian modeling in prior work.** We place our method within a unified view of existing exemplar-free, prototype-based continual learning methods (e.g., PASS, FeTrIL, FeCAM, EFC, AdaGauss, LDC, ADC), showing how they fit a common Gaussian/prototype framework and how our bidirectional transport extends this line.

**2. Appendix B.2: Measuring adherence to Gaussian assumptions.** We discuss the limitations of strict multivariate normality tests in high-dimensional deep feature spaces and provide empirical diagnostics indicating that class-conditional features exhibit approximately elliptical (Gaussian-like) geometry under the chosen backbone.

**3. Appendix B.3: t-SNE snapshots of class-conditional geometry.** We visualize the evolution of selected classes across tasks using t-SNE, overlaying Gaussian ellipses to show how well single Gaussians cover class clusters.

**4. Appendix B.4: Intuitive view and analysis of bidirectional cycle consistency.** We provide an intuitive explanation and spectral analysis of the bidirectional maps $(A, D)$ and the stop-gradient cycle loss, clarifying how they act as near-isometries between feature spaces and why this reduces long-term transport bias and prototype distortion.

**5. Appendix B.5: On AdaGauss, full-covariance prototypes, and robustness.** We clarify how AdaGauss models each class with a full-covariance Gaussian and uses a Cholesky-based anti-collapse loss, why dimensionality reduction (512→64) is necessary to make such modeling numerically feasible in exemplar-free CIL, and why this alone is still insufficient—motivating our robust variant that enforces SPD, uses shrinkage and jitter, and falls back to diagonal/eigenvalue-floored covariances when needed.

**6. Appendix B.6: Parameter and computational overhead.** We detail parameter counts and computational cost for the backbone plus projector(s) in the 64-dimensional setting, demonstrating that the bidirectional variant adds only a small overhead relative to AdaGauss while keeping the backbone, training schedule, and feature dimensionality identical across methods.

**7. Appendix B.7: Sensitivity to new hyperparameters.** We report a sensitivity study over the main hyperparameters $(\lambda_{\text{bi}}, \lambda_{\text{cyc}}, \alpha)$ on CIFAR-100, showing that our method consistently outperforms one-directional baselines across a reasonably wide range of values and does not rely on aggressive hyperparameter tuning.

**8. Appendix B.8: Compatibility with alternative classifiers.** We evaluate our bidirectional transport together with a standard fully connected “Linear (sampling)” head trained on Gaussian-rebalanced features, and find that it essentially matches the Gaussian Bayes classifier, indicating that our approach is not tied to a specific decision rule.

**9. Appendix B.9: Additional “training from half” (warm-start) results.** We add experiments in the training-from-half (warm-start) setting and analyze why EFC is favored when many head classes appear in the initial task, while our method remains superior in the more challenging cold-start regimes that are the focus of the main paper.

**10. Appendix B.10: Stress-test on long-tail continual dataset.** We further evaluate our method under the sequential long-tailed CIFAR-100-LT ($r=20$). This analysis shows that our bidirectional transport remains highly competitive under pronounced head–tail imbalance and that the warm-start behavior discussed in the main text extends to long-tailed streams, while also clarifying that fully general long-tailed EFCIL requires dedicated techniques that we leave for future work.

---

### Author Response · Authors · 2025-11-20
**Summary of changes 1/2**

We would like to sincerely thank all Reviewers for the time and effort invested in reading, analyzing, and commenting on our submission. Your detailed feedback, questions, and suggestions were instrumental in shaping this revised version. We carefully considered each point, introduced new experiments and analyses in both the main paper and the Rebuttal Appendix B, and we hope that the resulting manuscript better reflects the scope, strengths, and limitations of our approach and is now closer to your expectations.

**Major changes in the main paper.** The main paper has been revised as follows:

**1. Addition of DPCR baseline (ICML’25).** We extend our comparative study by including DPCR (He et al., ICML 2025) as a strong recent exemplar-free CIL baseline. Our method matches or outperforms DPCR across the used benchmarks.

**2. Prototype drift analysis with oracle prototypes.** We add a new drift analysis for the top-performing methods, where we compute the distance between maintained prototypes and “oracle” prototypes (empirical class means under the final backbone). This shows that our bidirectional transport consistently yields smaller prototype drift than prior work.

**3. Explicit parameter overhead calculation.** We explicitly quantify the parameter cost of the additional adapter in the 64-dimensional configuration, showing that it adds only a small fraction of the parameters of a ResNet-18 backbone and that the observed gains cannot be explained by model capacity alone.

**4. Improved clarity of Figure 2.** We refine Figure 2 and its accompanying explanation to more clearly illustrate the role of our propoesed method.

---

### Meta-Review · Area_Chair_ywjF · 2025-12-29

**Summary:**

The reviewers broadly agree that the paper addresses an important and well-motivated problem in exemplar-free class-incremental learning, namely prototype drift induced by one-directional projection schemes, and that the proposed bidirectional alignment with cycle consistency is conceptually sound and technically novel. Initial concerns focused on the reliance on Gaussian class modeling, the robustness of the method under fine-grained or low-drift regimes, the lack of recent baselines, insufficient clarity on parameter and computational overhead, and limited analysis of edge cases where gains are marginal. While some reviewers initially expressed reservations about experimental completeness and practical assumptions, the overall evaluation recognized the strong theoretical grounding, careful problem formulation, and extensive empirical validation across standard benchmarks.

**Reviewer Concerns:**

The rebuttal substantially addressed the major concerns raised during review. Questions regarding the Gaussian assumption were clarified through additional geometric analyses, visualizations, and stress tests, with a convincing argument that single-Gaussian prototypes serve as a robust geometric approximation rather than a strict generative model in the studied protocol. Concerns about outdated baselines, parameter overhead, hyperparameter sensitivity, warm-start settings, and classifier compatibility were all directly handled via newly added experiments and analyses, including the inclusion of DPCR, explicit parameter and compute breakdowns, sensitivity studies, training-from-half experiments, and evaluations with alternative classifiers.

**Reviewer Scores:**

Reviewer zrCc would likely stay at 8 given their already positive stance and the added empirical evidence and scope clarification around Gaussianity and fine-grained behavior. Reviewer J5qi would likely increase  because the rebuttal directly quantified overhead, added sensitivity analysis, clarified the stop-gradient cycle design with both intuition and theory, and discussed the regimes where gains are marginal. Reviewer tKrY explicitly indicated they would raise their score to 8 after the rebuttal fully addressed baseline freshness (DPCR), parameter sizing via the reduced feature space, hyperparameter sensitivity, warm-start results, and classifier compatibility. Reviewer A1WY would likely remain at 8 since their key requests—oracle prototype drift analysis, clearer inference description, improved Figure 2 clarity, and justification for transporting covariances—were all directly incorporated into the revision and rebuttal.

---

### Decision · Program_Chairs · 2026-01-26

Accept (Poster)